# FlashBias: Fast Computation of Attention with Bias

**Haixu Wu[1], Minghao Guo[2], Yuezhou Ma[1], Yuanxu Sun[1], Jianmin Wang[1],**
**Wojciech Matusik[2], Mingsheng Long[1]✉**
[1]School of Software, Tsinghua University, [2]MIT CSAIL
{wuhaixu98,guomh2014}@gmail.com, {mayz20,sunyuanx22}@mails.tsinghua.edu.cn,
{jimwang,mingsheng}@tsinghua.edu.cn, {wojciech}@csail.mit.edu

## Abstract

Attention with bias, which extends standard attention by introducing prior knowledge as an additive bias matrix to the query-key scores, has been widely deployed in vision, language, protein-folding and other advanced scientific models, underscoring its status as a key evolution of this foundational module. However, introducing bias terms creates a severe efficiency bottleneck in attention computation. It disrupts the tightly fused memory-compute pipeline that underlies the speed of accelerators like FlashAttention, thereby stripping away most of their performance gains and leaving biased attention computationally expensive. Surprisingly, despite its common usage, targeted efficiency optimization for attention with bias remains absent, which seriously hinders its application in complex tasks. Diving into the computation of FlashAttention, we prove that its optimal efficiency is determined by the rank of the attention weight matrix. Inspired by this theoretical result, this paper presents FlashBias based on the low-rank compressed sensing theory, which can provide fast-exact computation for many widely used attention biases and a fast-accurate approximation for biases in general formalizations. FlashBias can fully take advantage of the extremely optimized matrix multiplication operation in modern GPUs, achieving $1.5\times$ speedup for Pairformer in AlphaFold 3, and over $2\times$ speedup for attention with bias in vision and language models without loss of accuracy. Code is available at this repository: https://github.com/thuml/FlashBias.

## 1 Introduction

In recent years, Transformer [40] has enabled impressive achievements in extensive areas, including computer vision [25, 45], natural language processing [2, 39], scientific applications [1, 18, 42], etc. Especially, as the core design of Transformer, the attention mechanism with powerful relation modeling capacity has emerged as a foundation module in deep learning models, making its optimization of vital importance. As a seminal progress, FlashAttention [10, 9] speeds up the computation of standard attention by successfully reducing the read-write cost of GPU high bandwidth memory (HBM) with IO-aware techniques. Afterwards, researchers further extend FlashAttention to support diverse kinds of masks [35, 41], such as causal mask in decoder-only Transformers or sliding window mask.

While standard attention or attention with masks has enjoyed elaborative efficiency optimization, we notice that *attention with bias* is a similarly important extension in dealing with complex tasks, where useful prior knowledge is introduced as a bias term of dot-product attention weights to guide the model learning. For example, in vision and language Transformers, the relative distance among tokens is a well-established attention bias and has been proven essential to the final performance of various tasks [25, 24, 30]. Also, for scientific problems with rich domain-specific prior [42], attention bias is an indispensable component, such as the pair representation bias in AlphaFold [1, 18, 34]. However, the targeted efficiency optimization for attention with bias is still lacking. All previous research in extending standard attention is centered on the speedup of attention with masks [35, 41], where the

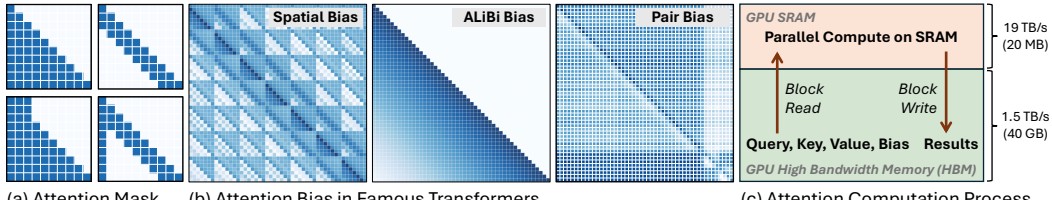

(a) Attention Mask    (b) Attention Bias in Famous Transformers    (c) Attention Computation Process

Figure 1: (a-b) Comparison between attention mask and bias, where the spatial bias is from Swin Transformer [25] for computer vision, ALiBi bias is used in language modeling [30] and pair bias is from AlphaFold [1]. (c) FlashAttention needs to read bias tensors from HBM to the on-chip SRAM.

sparsity nature of masks enables possible computation reduction. However, unlike the attention mask that defines computation logic, the bias matrix describes the pairwise relation among tokens, which is inherently continuous and dense as showcased in Figure 1, making previous sparsity-based speedup techniques inapplicable. Although FlexAttention [11], benefiting from compiler techniques, a new feature in PyTorch 2.5 [27], can support general formalizations of bias terms, it still depends on element-wise operations that are less optimized than matrix multiplications and fails in speeding up dynamic bias. To date, the fast computation of attention with bias remains a nascent area to explore.

In FlashAttention [10], researchers find that instead of computation FLOPS, the read-write (IO) overload of GPU high bandwidth memory (HBM) is the actual bottleneck of speed. As the bias matrix is usually dense, it is really hard to bypass the quadratic IO complexity, where the computation needs to read the whole bias term from HBM at least once, making its speedup intractable. In this paper, we notice that the IO challenge that we face here can be recast into the classical compressed sensing problem [12], whose basic assumption is that the "measurement" (corresponding to IO overload here) is expensive but the computation (corresponding to the fast on-chip computation) is cheap. This new perspective offers us valuable theoretical understandings. Specifically, for a dense matrix, the optimal "measurement" (IO overload) is highly related to the matrix's rank [6]. This inspires us to dive into the rank of attention bias and surprisingly find that most of the widely used biases are inherently of low rank, eventually discovering one flexible way to enable fast computation of attention with bias.

Based on the above understandings, this paper presents FlashBias based on the low-rank compressed sensing theory. By formalizing commonly used attention biases into the multiplication of two factor tensors, FlashBias achieves fast and exact computation for many Transformers, covering vision, language and scientific domains. Going further, we present an approximation method for general bias formalization with a low-rank neural decomposition technique, which successfully speeds up more complicated attentions in AlphaFold 3 [1]. Our contributions are summarized as follows:

- Based on an in-depth study of the computation bottleneck of attention with bias, this paper theoretically proves that for a dense matrix, e.g. dot-product attention weights or various attention biases, the optimal efficiency in GPU is determined by the rank of the matrix.

- Inspired by theoretical analyses, this paper presents FlashBias based on the low-rank compressed sensing theory, which utilizes exact and SVD decompositions for fast computation of many widely-used attention biases, and an approximation version for general biases.

- FlashBias can accelerate a family of widely-used backbones without loss of accuracy, which brings $1.5\times$ speedup for AlphaFold and over $2\times$ speedup for vision and language models.

## 2 Preliminaries

### 2.1 Attention with Bias

Attention [40] contains queries $\mathbf{q} \in \mathbb{R}^{N \times C}$, keys $\mathbf{k} \in \mathbb{R}^{M \times C}$ and values $\mathbf{v} \in \mathbb{R}^{M \times C}$, where $N, M$ denote the sequence length of queries and keys respectively and $C$ represents the channel of hidden representations. Conventionally, attention weights are calculated based on the dot-product between queries and keys. Beyond solely relying on the representation dot-product, useful prior knowledge is also commonly introduced into the attention mechanism as a bias term to guide learning, which is:

$$\mathbf{o} = \text{softmax}(\frac{\mathbf{q}\mathbf{k}^\top}{\sqrt{C}} + \mathbf{b})\mathbf{v}. \tag{1}$$

Here $\mathbf{o} \in \mathbb{R}^{N \times C}$ denotes results and $\mathbf{b} \in \mathbb{R}^{N \times M}$ represents the prior weight for query-key pairs, which is also referred to as *attention bias* [18, 25]. Actually, attention bias has been widely used in diverse domains and has been proven to be indispensable for model learning, especially for complex tasks. For example, in computer vision, Swin Transformer [25] adopts the relative distance among different pixels as attention biases to introduce the essential locality inductive bias into vision backbones. Similarly, relative position is also used to encode sequential information in language models [14, 39]. In addition, attention bias widely exists in Transformer-based scientific deep models to incorporate domain-specific priors, such as the pair representation bias in AlphaFold [1, 18, 34].

It is worth noting that *attention mask* is also commonly used in attention mechanisms to enable the calculation under a specific logic, such as the upper triangle mask (aka. causal mask) in decoder-only Transformers [2] and sliding window mask in sparse Transformers [3]. Usually, the mask is also applied as an additive term to the query-key dot-product. However, attention biases that we focus on in this paper are usually dense continuous values that incorporate detailed prior knowledge into each query-key pair to reflect the degree of token relations, while masks only contain zeros and negative infinite values and are usually sparse. This difference makes their speed-up strategies distinct.

Additionally, large language models widely adopt rotary position embedding (RoPE) [37] to introduce relative position into attention, which can be considered as a special multiplicative bias and is already computation efficient. Considering that RoPE is outside the additive bias scope, we defer its discussion to Appendix J and will showcase that FlashBias can also be extended to multiplicative bias.

## 2.2 Fast Computation of Attention

As the core component in modern deep models [2, 18, 25, 29], the attention mechanism has been widely explored. Although it has demonstrated impressive performance, one serious drawback is its computational complexity, which is quadratic w.r.t. sequence length. To tackle the efficiency bottleneck, the fast and exact computation of the attention mechanism has also been widely investigated. Based on tiling techniques, researchers formalized the softmax function in attention with a sequential calculation process [26, 31], successfully reducing memory cost to linear complexity. Afterwards, FlashAttention [10] and FlashAttention-2 [9] further demystify that the IO overload of GPU HBM is the actual efficiency bottleneck and enable significant speedup by utilizing the super-fast on-chip SRAM. The FlashAttention family has served as a default acceleration option of modern Transformers and has been built-in and supported by PyTorch [27].

Further, to extend the model's capability in handling diverse types of attention, researchers have made a great effort to enable fast computation for attention with masks, such as the causal mask [2] and sliding window mask [3], etc. Technically, the design principle in speeding up attention with masks is to utilize the sparsity in masks to reduce the IO complexity. For instance, Binary Block Masking [35] splits the mask matrix as blocks and adopts binary values to indicate the blocks with non-zero masks, which enables the masking process to skip the IO read of all-zero mask blocks. Subsequently, FlashMask [41] utilizes the spatial continuity of non-zero mask values and proposes to only record the start and end indices of mask segments, achieving fast computation under rich mask types. Although the above works can speed up a wide range of mask types, as attention bias is usually continuous and dense, these sparsity-based methods cannot be applied to attention bias.

Recently, FlexAttention [11], taking advantage of deep learning compiler techniques, has been released in PyTorch 2.5, which supports a wide range of masks (e.g. causal mask [2]) and biases (e.g. ALiBi [30]) by pre-compiling element-wise operations. However, since many on-chip calculations are less optimized than matrix multiplications, FlexAttention cannot achieve a perfect speedup and fails to support data-dependent dynamic biases. In contrast, FlashBias can take full advantage of extremely optimized matrix multiplications and is also applicable to complex dynamic biases.

## 3 Method

As aforementioned, attention with bias is essential for model learning, while its natively dense and quadratic tensor shape results in a serious IO bottleneck. Inspired by the compressed sensing theory [12, 6], we verify that many well-established attention biases are inherently of low rank, paving the way for fast computation of attention with bias. Specifically, we dive into the computation of FlashAttention and theoretically prove that the optimal efficiency is inherently determined by the rank of attention weights and bias terms. Further, we present FlashBias based on the low-rank decomposition, which achieves optimal efficiency with natural adaptation for on-chip computing.

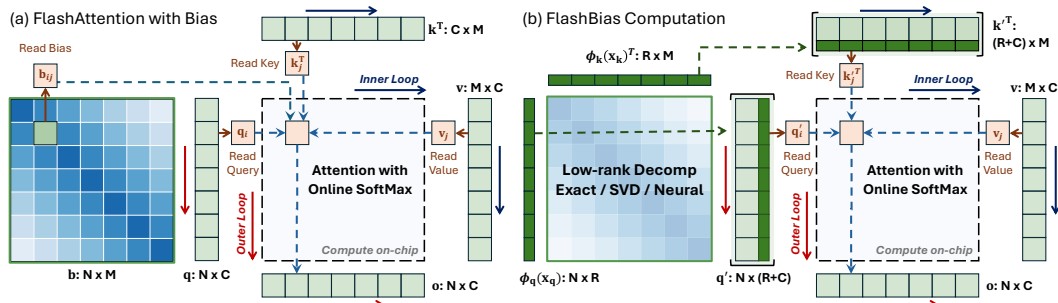

Figure 2: Comparison between FlashAttention and FlashBias. FlashBias utilizes low-rank decomposition to bypass the read of the whole bias matrix, successfully avoiding the quadratic IO overload.

## 3.1 Rethinking FlashAttention Computation

We begin by analyzing the theoretical basis of FlashAttention's speedup (without bias or mask) on dense and continuous attention weights. Our analysis yields that the rank of dense matrices, such as the dot-product attention weight $\mathbf{s} = \mathbf{q}\mathbf{k}^\top \in \mathbb{R}^{N \times M}$ and the bias matrix $\mathbf{b} \in \mathbb{R}^{N \times M}$, inherently decides the IO cost, which is formally stated as follows. All the proofs can be found in Appendix A.

**Theorem 3.1** (FlashAttention computation benefits from low rank). *Suppose $N = M$ and let $R$ be the rank of dot-product attention weight $\mathbf{s}$, $C = \alpha N$ be the channel dimension with constant $\alpha$ and sequence length $N$, $S$ be the size of SRAM with $S = \beta N C$ and $\frac{1}{N} \leq \beta \leq 1$. Then, 1) the HBM access of FlashAttention is $\Theta\left((1 + \frac{1}{\alpha})\beta\right)$ times smaller than the standard attention, and 2) $\alpha \geq \frac{R}{N}$.*

As demonstrated in Theorem 3.1, the speedup of FlashAttention is proportional to $\beta$ (SRAM size) and inversely proportional to $\alpha$ (channel dimension). If we consider the attention weight $\mathbf{s}$ as a low-rank matrix, the optimal speedup of FlashAttention is obtained by reducing the channel dimension to $R$, i.e. $\alpha = \frac{R}{N}$. The same technique is also used in DeepSeek-v3 [23] as Multi-Head Latent Attention, which reduces the channel dimension by projecting $\mathbf{q}, \mathbf{k}, \mathbf{v}$ into a small latent space for acceleration.

**Theorem 3.2** (Compressed sensing complexity of low-rank dense matrix [6]). *Given a $N \times N$ dense matrix with rank $R$, the theoretically optimal compressed tensor is of storage complexity $\Theta(NR)$.*

Theorem 3.2 demonstrates that the optimal storage of bias is linearly related to its rank, highlighting the essentiality of the low-rank property. Further, integrating with prior analyses of HBM access in exact attention [9], we can derive the IO complexity property of HBM access in attention with bias.

**Corollary 3.3** (A "lower bound" for HBM access of attention with bias). *Given $\mathbf{q} \in \mathbb{R}^{N \times C}, \mathbf{k}, \mathbf{v} \in \mathbb{R}^{M \times C}$, bias $\mathbf{b}$ of rank $R$ and SRAM of size $S$ where $(C + R) \leq S \leq N(C + R)$, there does not exist an algorithm to compute exact attention with bias through $o\left(\frac{NM(C^2 + R^2)}{S}\right)$ HBM access for all $S \in [(C + R), N(C + R)]$. Here $o(*)$ represents the strict asymptotic upper bound.*

## 3.2 FlashBias

Inspired by the above theoretical results, we present FlashBias based on low-rank decomposition techniques, with novel design to utilize the low-rank property of attention bias to reduce HBM access.

**Overall design** As shown in Figure 2, instead of block-wise reading bias, FlashBias replaces the quadratic bias matrix $\mathbf{b} \in \mathbb{R}^{N \times M}$ as two factor tensors. Specifically, considering a bias matrix calculated by $\mathbf{b} = f(\mathbf{x_q}, \mathbf{x_k})$, where $\mathbf{x_q} \in \mathbb{R}^{N \times C'}, \mathbf{x_k} \in \mathbb{R}^{M \times C'}$ represent the source information for generating the bias, which is set as spatial position of each pixel in Swin Transformer [25] and the representation of protein residues in AlphaFold [1], if there exist factor functions $\phi_\mathbf{q}, \phi_\mathbf{k}$ satisfying:

$$f(\mathbf{x_q}, \mathbf{x_k}) = \phi_\mathbf{q}(\mathbf{x_q})\phi_\mathbf{k}(\mathbf{x_k})^\top, \quad \phi_\mathbf{q}, \phi_\mathbf{k} : \mathbb{R}^{C'} \to \mathbb{R}^R. \tag{2}$$

The computation of attention with bias can be equivalently formalized as follows:

$$\mathbf{o} = \text{softmax}(\frac{\mathbf{q}\mathbf{k}^\top}{\sqrt{C}} + \mathbf{b})\mathbf{v} = \text{softmax}\left(\frac{[\mathbf{q}|\sqrt{C}\phi_\mathbf{q}(\mathbf{x_q})][\mathbf{k}|\phi_\mathbf{k}(\mathbf{x_k})]^\top}{\sqrt{C}}\right)\mathbf{v}, \tag{3}$$

Table 1: The computation of FlashBias for widely-used attention biases, which includes three different types: (a) Exact decomposition by finding exact $\phi_{\mathbf{q}}, \phi_{\mathbf{k}}$, (b) SVD decomposition for cases using model parameter as bias, (c) Neural decomposition for using model representation as dynamic bias.

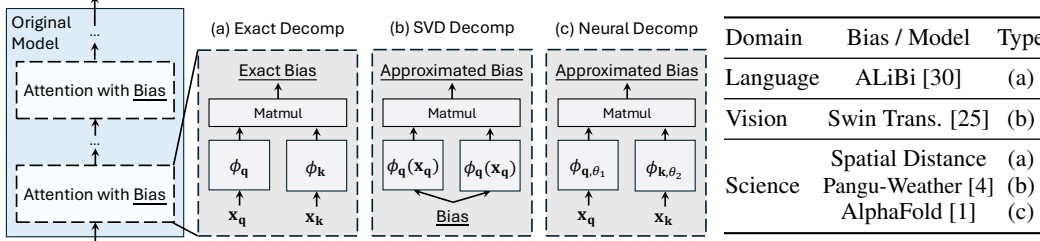

| Domain | Bias / Model | Type |
|---|---|---|
| Language | ALiBi [30] | (a) |
| Vision | Swin Trans. [25] | (b) |
| Science | Spatial Distance | (a) |
| | Pangu-Weather [4] | (b) |
| | AlphaFold [1] | (c) |

where $[*|*]$ denotes the concatenation operation along the channel dimension. Notably, this design significantly reduces the storage cost for the attention bias from $\mathcal{O}(NM)$ to $\mathcal{O}\left((N+M)R\right)$. Although its design will require recalculating the bias weight, this computation is just a simple matrix multiplication of $\phi_{\mathbf{q}}(\mathbf{x_q})\phi_{\mathbf{k}}(\mathbf{x_k})^{\top}$, an operation that has been extremely optimized on modern GPUs.

Such a simple design is broadly applicable to a wide range of variants for attention with bias. In practice, we implement it through three concrete instantiations for $\phi_{\mathbf{q}}, \phi_{\mathbf{k}}$, as shown in Table 1.

**Exact decomposition**  We find that some well-established attention biases can be directly decomposed into factor functions, enabling fast and exact computation. Here are representative cases.

**Example 3.4** (ALiBi [30] in language models). *Given $\mathbf{x_q} = [1, \cdots, N], \mathbf{x_k} = [1, \cdots, M]$, the ALiBi bias is calculated as $f(\mathbf{x}_{\mathbf{q},i}, \mathbf{x}_{\mathbf{k},j}) = i - j$, which can be directly decomposed into a low-rank formalization by defining $\phi_{\mathbf{q}}(\mathbf{x}_{\mathbf{q},i}) = [1, i]$ and $\phi_{\mathbf{k}}(\mathbf{x}_{\mathbf{k},j}) = [-j, 1]$, corresponding to the case $R = 2$. The original ALiBi also involves a causal mask, while we only focus on the bias term here.*

**Example 3.5** (Spatial distance in scientific problems). *Transformers has been used as surrogate models for PDE solving [43], especially for aerodynamic simulation. It is critical to introduce spatial distance to guide attention learning among massive computational points. Let $\mathbf{x_q} = \mathbf{x_k} \in \mathbb{R}^{N \times 3}$ record the 3D spatial positions of $N$ computation points, where $\mathbf{x}_{\mathbf{q},i} \in \mathbb{R}^3$ is the position of $i$-th point. For the spatial distance $f(\mathbf{x}_{\mathbf{q},i}, \mathbf{x}_{\mathbf{k},j}) = \|\mathbf{x}_{\mathbf{q},i} - \mathbf{x}_{\mathbf{k},j}\|_2^2$, it can be exactly decomposed as:*

$$\phi_{\mathbf{q}}(\mathbf{x}_{\mathbf{q},i}) = [\mathbf{x}_{\mathbf{q},i,0}^2, 1, -2\mathbf{x}_{\mathbf{q},i,0}, \mathbf{x}_{\mathbf{q},i,1}^2, 1, -2\mathbf{x}_{\mathbf{q},i,1}, \mathbf{x}_{\mathbf{q},i,2}^2, 1, -2\mathbf{x}_{\mathbf{q},i,2}],$$
$$\phi_{\mathbf{k}}(\mathbf{x}_{\mathbf{k},j}) = [1, \mathbf{x}_{\mathbf{k},j,0}^2, \mathbf{x}_{\mathbf{k},j,0}, 1, \mathbf{x}_{\mathbf{k},j,1}^2, \mathbf{x}_{\mathbf{k},j,1}, 1, \mathbf{x}_{\mathbf{k},j,2}^2, \mathbf{x}_{\mathbf{k},j,2}]. \tag{4}$$

**SVD decomposition**  Some models such as Swin Transformer [25] and Pangu-Weather [4] adopt the learnable model parameters for relative position encoding. Specifically, each bias term in their model is an $N \times M$ matrix of model parameters. As this type of bias is fixed once the model has been well trained, it is convenient to conduct Singular Value Decomposition (SVD) [20] for low-rank decomposition of these parameters. In practice, we precompute SVD once offline, incurring negligible runtime overhead. The resulting decomposed factor tensors can then be utilized to accelerate the subsequent inference process, thanks to their low-rank nature.

**Neural decomposition**  The two mechanisms discussed above apply to static bias terms. Beyond these, some models introduce more complex or dynamically generated biases. For example, in AlphaFold [1], the bias term is projected from the intermediate pair representations. In this case, its low-rank decomposition cannot be explicitly or exactly derived. Also, due to the data-dependent property, SVD decomposition needs to be conducted for bias at every inference. To accelerate these complex biases, we present a neural approximation version of FlashBias, which employs two lightweight neural networks $\widehat{\phi}_{\mathbf{q},\theta_1}, \widehat{\phi}_{\mathbf{k},\theta_2} : \mathbb{R}^{C'} \to \mathbb{R}^R$ to approximate factor functions $\phi_{\mathbf{q}}$ and $\phi_{\mathbf{k}}$, which is supervised by the following objective function:

$$\min_{\theta_1, \theta_2} \mathcal{L}(\mathbf{x_q}, \mathbf{x_k}) = \|\widehat{\phi}_{\mathbf{q},\theta_1}(\mathbf{x_q})\widehat{\phi}_{\mathbf{k},\theta_2}(\mathbf{x_k})^{\top} - f(\mathbf{x_q}, \mathbf{x_k})\|_2^2. \tag{5}$$

Here $\theta_1, \theta_2$ are learnable parameters, which can be optimized by fine-tuning $\theta_1, \theta_2$ on the training set. Note that FlashBias attempts to completely replace the original bias term, a design fundamentally different from LoRA [16] which learns an additive term to the pretrained model parameters. Similar to the SVD decomposition version, once these two lightweight neural networks $\widehat{\phi}_{\mathbf{q},\theta_1}, \widehat{\phi}_{\mathbf{k},\theta_2}$ have been well optimized, they can be directly applied to all future inference.

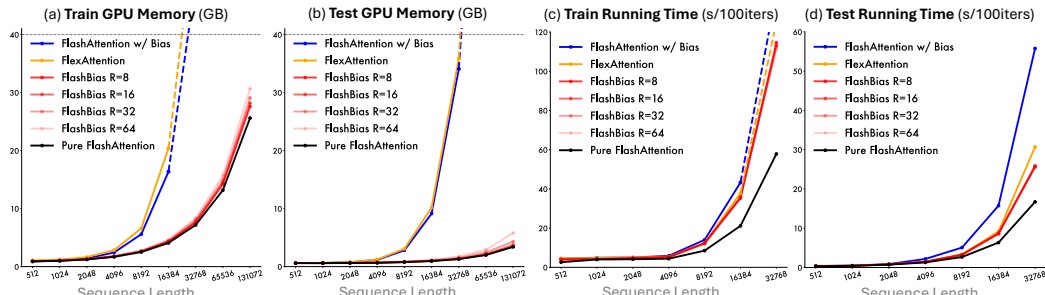

Figure 3: Efficiency comparison among FlashBias, FlashAttention w/ Bias and FlexAttention [11]. Here "Pure FlashAttention" refers to canonical FlashAttention without a bias term, which can be viewed as an upper bound of computation efficiency. Dotted lines indicate out-of-memory situations.

**Remark 3.6** (Understanding neural decomposition). *In Eq. (5), we define factor networks $\widehat{\phi}_{\mathbf{q},\theta_1}, \widehat{\phi}_{\mathbf{k},\theta_2}$ as token-wise functions, instead of depending on the whole sequence. This is because elements in bias that reflect pair-wise relations are solely determined by corresponding token information, such as token position in relative position bias. Besides, rank $R$ essentially controls the dimension of dot-product space, thereby should be set based on "complexity" of prior knowledge that derives bias.*

**Corollary 3.7** (HBM access of FlashBias). *Given $\mathbf{q} \in \mathbb{R}^{N \times C}, \mathbf{k}, \mathbf{v} \in \mathbb{R}^{M \times C}$, rank-$R$ bias $\mathbf{b}$ and size-$S$ SRAM ($C \leq S \leq NC$), the HBM access complexity of FlashBias is $\Theta\left(\frac{NM(C^2+R^2)}{S}\right)$.*

**Remark 3.8** (Trade-off between approximation accuracy and efficiency). *Although SVD and neural decomposition will introduce approximation error, owing to the native low-rank property of attention bias in modern Transformers, it will not affect the final performance by setting $R$ as a reasonable value. For example, in some layers in Swin Transformer, $R = 32$ can maintain over 99% energy of the original bias with shape $576 \times 576$. More case studies are included in Appendix E and F.*

**Example 3.9** (Comparison with FlashAttention). *For FlashAttention with bias, its HBM access is $\Theta\left(\frac{NMC^2}{S} + NM\right)$ [9]. Given a regular setting of Transformers and GPU ($C = 64$, $S$ is 100KB) and supposing $R = 64$, $N, M \gg C, R$, then HBM IO of FlashBias is $\approx 6\times$ smaller than FlashAttention.*

**Speed up training**    In FlashBias, the exact decomposition version can be directly applied to both training and inference phases. As for SVD or neural decomposition, their computations are based on a pretrained model; thereby, the inference speedup is obvious. Note that for large-scale pretrained models, such as AlphaFold [1], the inference speedup is already sufficiently valuable. Going further, it is also possible to speed up the training phase with these two types of biases. Specifically, for the SVD type, we can replace the $N \times M$ model parameter bias with $N \times R$ and $M \times R$ model parameter tensors at the initialization phase. As for the neural type, instead of using the whole bias matrix, it can also be replaced with two lightweight neural networks when defining the model architecture.

## 4    Experiments

As summarized in Table 2, we extensively test FlashBias on Transformers for language modeling, computer vision and scientific problems, which can significantly speed up model efficiency at both training and test phases. This section will first present an overall efficiency comparison among different techniques and then integrate FlashBias to different domain-specific Transformers with various bias types.

Table 2: Summary of experiments and base models.

| Base Model | Bias Type | Phase |
|---|---|---|
| Plain Transformer [40] | Static | Train & Inference |
| GPT-2 [32] | Static | Train & Inference |
| Swin Trans. [24] | Learnable | Inference |
| PDE Solver [43] | Learnable | Train & Inference |
| AlphaFold 3 [1] | Learnable | Inference |

All experiments were performed in PyTorch 2.5.0 [27] and Triton 3.0.0 [38] on a single A100 GPU.

### 4.1    Overall Comparison

**Setups**    To give a clear comparison among FlashBias, vanilla FlashAttention with Bias [9] and the latest FlexAttention [11], we make a comprehensive efficiency evaluation based on a plain

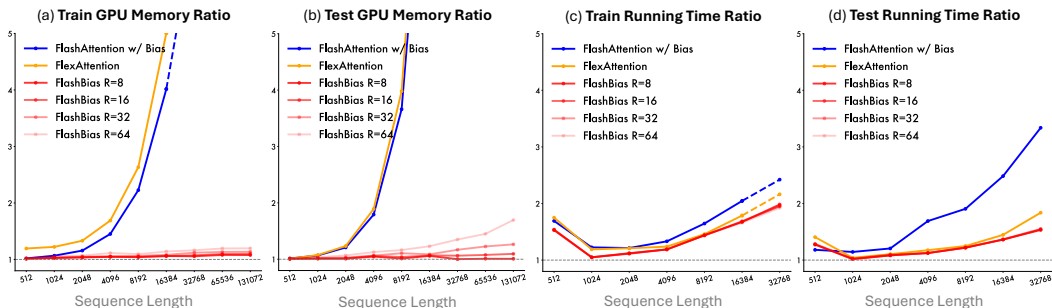

Figure 4: Efficiency ratio over "Pure FlashAttention", which is calculated by $\frac{\text{Method Efficiency}}{\text{Pure FlashAttention Efficiency}}$.

Transformer [40], which consists of 8 layers. Each Transformer layer involves a feedforward network with 1024 intermediate hidden channels and attention with 512 hidden channels, 8 heads, as well as a static bias matrix of shape $\#\,\text{heads} \times N \times N$. All the metrics are recorded with a batch size of 1.

**Results**    As shown in Figure 3 and 4, we can find that FlashBias (red lines) consistently outperforms FlashAttention with Bias [9] and FlexAttention [11], demonstrating the effectiveness of our design.

According to Figure 3(a-b), when sequence length $N = 16384$, FlashBias can significantly reduce GPU memory usage, which is $5\times$ smaller than FlexAttention and FlashAttention with Bias during training and $10\times$ smaller during inference. This memory benefit can also be theoretically justified by Theorem 3.2. As for training and inference running time, FlashBias presents 18.6% and 44% speedup compared to vanilla usage of FlashAttention and is also better than FlexAttention in long sequences.

It is worth noticing that, although FlexAttention [11], which optimizes FlashAttention with deep learning compiler techniques, can significantly boost the running time compared to vanilla FlashAttention with Bias (Figure 4 (c-d) orange lines), it still cannot reduce GPU memory due to the quadratic storage cost of the bias matrix ((a-b) orange lines). In addition, due to the HBM access overload of the bias matrix, it falls short in running time when inputting long sequences. These observations further demonstrate that the low-rank property utilized in FlashBias is the key to efficiency.

**Implementation choices**    As formalized in Eq. (3), for implementation, researchers can concatenate the decomposed factor functions with query and key tensors in the original attention along the channel dimension, which means FlashBias can seamlessly adopt previous efficient implementations for canonical attention, such as PyTorch SDPA[1]. In pursuit of a well-optimized operation, we also implement FlashBias based on the Triton backend [38]. Figure 5 demonstrates that the Triton implementation is much more efficient in the forward pass and the PyTorch SDPA-based version is better in the training phase. Therefore, we adopt the Triton version for inference and PyTorch SDPA version for

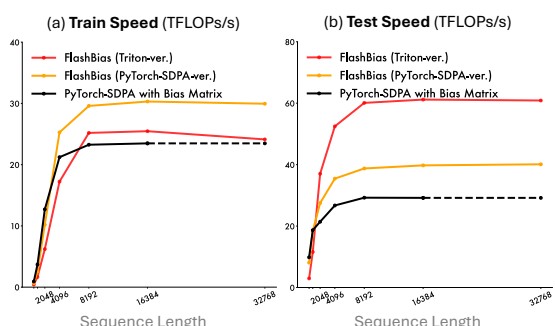

Figure 5: Speed of different implementations on attention with 128 hidden channels and 8 heads. $R$ is set as 8 in FlashBias. Dotted lines indicate out-of-memory. More attention speed comparisons are in Appendix B.

training throughout this paper. Besides, although vanilla PyTorch SDPA can accelerate the computation, it still suffers from the out-of-memory issue, which can be perfectly resolved by FlashBias.

## 4.2   Large Language Model

**Setups**    Here we employ FlashBias to speed up ALiBi [30], which is a well-known bias in language modeling and has been proven better than rotary position embedding [37] in handling input length extrapolation. We follow the configuration in GPT-2 [32] and evaluate based on a large language model with ALiBi bias, which contains 48 decoder-only Transformer layers, 1.5B parameters in total.

---

[1]PyTorch implementation of `torch.nn.functional.scaled_dot_product_attention`

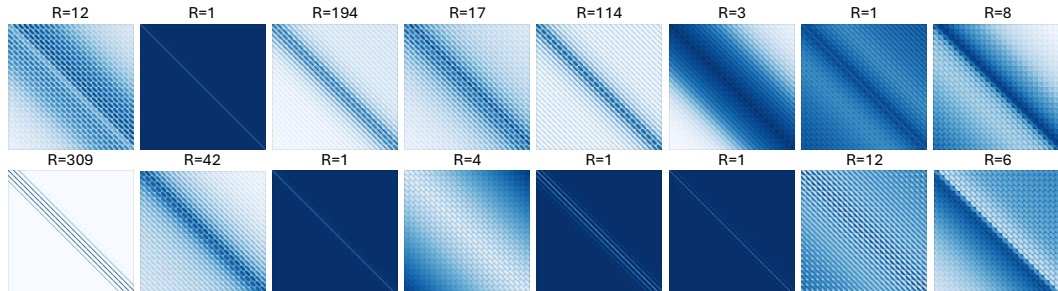

Figure 6: Visualization of bias matrix in SwinV2-B. $R$ can maintain 99.5% energy of the bias matrix.

Each layer contains attention with 1600 channels and 50 heads, as well as a feedforward network with 6400 hidden dimensions. Beyond large model size, this task also involves the causal mask.

**Implementation**   As stated in Example 3.4, we can adopt the exact decomposition for ALiBi bias, where $R$ is equal to 2 and the result of FlashBias is exactly equivalent to the original computation. Since attention masking has been well-optimized in FlashAttention [9] and FlexAttention [11], we directly integrate them with our method to support the decoder-only Transformer. This combination also demonstrates that FlashBias is orthogonal to attention-masking speed-up techniques. Notably, FlashAttention has an `ALiBi_slopes` feature, which does not load bias matrix from HBM but computes the bias weights through just-in-time compilation. Since this is not a generic technique, we do not employ this feature in the main text. More discussion can be found in Appendix D.

**Results**   As mentioned above, since this task requires causal attention, we test FlashBias by integrating it with the attention mask speedup techniques in FlashAttention and FlexAttention. Results in Table 3 demonstrate that FlashBias still outperforms FlashAttention and FlexAttention in processing the ALiBi bias term. Specifically, in processing bias, FlashBias reduces over 50% (5.0→2.3) time cost of FlashAttention and over 25% (3.4→2.5) time cost of FlexAttention.

Table 3: Experiment of GPT-2. #Time records the running time for 100 iterations when $N = 2048$. $\Delta$ refers to the time difference w.r.t. Pure Causal *Attention without bias, reflecting the additional cost in processing the bias matrix.

| Method | Training | | Inference | |
|---|---|---|---|---|
| | Time(s) | $\Delta$ | Time(s) | $\Delta$ |
| Pure Causal FlashAttention | 119.3 | - | 38.77 | - |
| FlashAttention with Bias | 124.3 | 5.0 | 40.32 | 1.55 |
| **FlashBias (Ours)** | **121.6** | **2.3** | **39.26** | **0.49** |
| Pure Causal FlexAttention | 119.0 | - | 38.76 | - |
| FlexAttention with Bias | 122.4 | 3.4 | 40.03 | 1.27 |
| **FlashBias (Ours)** | **121.5** | **2.5** | **39.78** | **1.02** |

Note that ALiBi bias is relatively simple, enabling the compiler-based FlexAttention with a favorable speedup. However, it may degenerate in handling more complex bias matrices, such as the bias in Swin Transformer [24] in the next section.

## 4.3   Vision Transformer

**Setups**   We test FlashBias on the image classification task based on Swin Transformer v2 [24]. Specifically, we adopt the open-source SwinV2-B model[2]. This model contains 24 layers with input resolution as $384 \times 384$ and window size as $24 \times 24$, thus, the sequence length of its WindowAttention is 576. WindowAttention in every layer contains a relative position bias with size $\#\text{heads} \times 576 \times 576$, which is set as a learnable model parameter. We attempt to speed up the WindowAttention computation with FlashBias. All the efficiency metrics are evaluated under a batch size of 64.

**Implementation**   Since the biases are set as learnable parameters, we directly obtain them from a pretrained model and adopt the SVD decomposition to generate decomposed factor tensors. However, we observed that in one layer, not all heads present low-rank bias matrices, as shown in Figure 6. Thus, we only adopt FlashBias to layers where most of heads' biases are of low rank, that is, the last 8 layers of SwinV2-B, while employing vanilla FlashAttention with Bias for the other layers.

**Results**   In Table 4, we can find a serious performance drop in "Pure FlashAttention" (without bias, Acc@1 87%→9%), which verifies the essentiality of bias and also highlights the importance of

---

[2]Model and training configurations of SwinV2-B.

speedup attention with bias. Further, compared to the official implementation, FlashBias can reduce 60% running time and 27% memory cost, which is valuable for real-time development.

More importantly, FlashBias will not affect the performance, where the top-5 accuracy drops less than 0.02% and the fluctuation of the top-1 accuracy (+0.042%) is within the standard deviation. In particular, even using well-established quantization techniques will still cause a 0.64% top-1 accuracy drop for 22% speedup according to the FasterTransformer document[3]. This comparison demonstrates that the low-rank property is a principal basis for fast computation of attention.

Table 4: Experiment of SwinV2-B on ImageNet-1K. #Time and #Mem correspond to inference efficiency on A100 per batch. Offline calculation of SVD for all biases takes 4.79s.

| Method | Acc@1 | Acc@5 | Time(s) | Mem(MB) |
|---|---|---|---|---|
| Official Code | 87.144% | 98.232% | 0.473 | 12829 |
| Pure FlashAttention | 9.376% | 19.234% | 0.180 | 3957 |
| FlashAttention with Bias | 87.142% | 98.232% | 0.230 | 11448 |
| FlexAttention [11] | 87.142% | 98.232% | 2.885 | 25986 |
| INT8 PTQ | 86.46% | | Around 22% speed up | |
| **FlashBias (Ours)** | 87.186% | 98.220% | 0.190 | 9429 |

Also, FlexAttention seriously degenerates in SwinV2-B. This is because, unlike experiments in Section 4.1, Swin Transformer's bias matrices are different in value and shape among different layers, which requires recompilation each time. This issue has also been mentioned by FlexAttention's author as "*If you are adding a [B, H, N, N] bias tensor, then you honestly shouldn't be using FlexAttention*"[4].

## 4.4 Scientific Deep Models

In addition to language and vision tasks, scientific problems usually involve rich domain-specific prior knowledge; thereby, attention bias also widely exists in scientific Transformers. Here we evaluate FlashBias in two representative models: Transformer-based PDE solvers [43] and AlphaFold 3 [1].

**Transformer-based PDE solvers**   Attention mechanism has been proven equivalent to a Monte-Carlo approximation of the integral operator [21], justifying its theoretical foundation for solving partial differential equations (PDEs). However, in processing complex geometries, the attention mechanism may fall in perceiving the 3D space, encouraging the utilization of spatial distance prior. Here we follow the driving car simulation task in [43], whose input is the position of computation mesh points and output is the physics quantities on these points. We test FlashBias on an 8-layer Transformer solver with a 3D distance bias described in Example 3.5. Each layer contains attention with 128 hidden dimensions and 8 heads, as well as a feedforward network with 256 hidden channels.

To approximate the adaptive mesh in numerical methods [36], we include a token-wise learnable weight $\alpha_i$ for the 3D distance bias in each head of every layer, i.e. $f(\mathbf{x}_{\mathbf{q},i}, \mathbf{x}_{\mathbf{k},j}) = \alpha_i \|\mathbf{x}_{\mathbf{q},i} - \mathbf{x}_{\mathbf{k},j}\|_2^2$. Unlike bias discussed in ALiBi [30] or SwinV2 [25], the learnable weights require the training process to record the gradient of the bias matrix, posing a challenging efficiency issue in backpropagation.

Table 5: Experiments of Transformer PDE solvers. The efficiency metrics are recorded under a batch size of 1 in the format of *#Mem (GB) / #Time (s/100iters)*. Accuracy comparisons are in Appendix G.

| Method (Learnable Bias) | Training Phase | | | Inference Phase | | |
|---|---|---|---|---|---|---|
| | 8192 | 16384 | 32186 | 8192 | 16384 | 32186 |
| FlashAttention | 12.8 / 15.4 | OOM | OOM | 4.54 / 5.46 | 15.3 / 21.2 | OOM |
| FlexAttention | *Not supported in current version* | | | 21.9 / 184.0 | OOM | OOM |
| **FlashBias (Ours)** | **1.46 / 4.54** | **2.02 / 14.7** | **2.97 / 51.1** | **0.98 / 1.22** | **1.03 / 3.48** | **1.13 / 12.7** |

**Results**   FlashBias is the only method that can support training of the Transformer solver on 32186 points (Table 5), which also presents a significant memory and running time advantage compared with other methods. Notably, FlashAttention and FlexAttention cannot support learnable bias training well, as they need to record dense gradient matrices, further highlighting the practicality of FlashBias.

**AlphaFold 3 for protein folding**   As a significant progress of AI for science, AlphaFold 3 [1] employs an elaborate architecture. Specifically, its core design, Pairformer, contains 144 attention

---

[3]FasterTransformer released by NVIDIA.
[4]Discussion about FlexAttention with dynamic bias matrix.

Table 6: Experiment of AlphaFold 3. The left part illustrates folding examples. Note that AlphaFold 3 is based on a diffusion model; thereby, slight differences are normal. The right table's efficiency is tested on the one-time inference of Pairformer with 1218 protein residue tokens (PDB ID: 7wux).

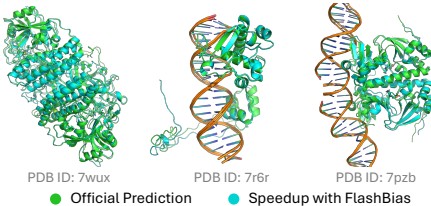

PDB ID: 7wux    PDB ID: 7r6r    PDB ID: 7pzb
● Official Prediction    ● Speedup with FlashBias

| Method | Test Set | | PDB ID 7wux | |
| | pLLDDT Loss ↓ | pTM ↑ | Time(s) | Mem(GB) |
|---|---|---|---|---|
| Open-sourced Code | 3.3724 | 0.9500 | 26.85 | 13.62 |
| FlashAttention w/o Bias | 4.3669 | 0.1713 | 8.27 | 12.89 |
| FlashAttention w/ Bias | 3.3724 | 0.9500 | 20.39 | 13.62 |
| **FlashBias (Ours)** | 3.3758 | 0.9498 | 18.19 | 13.62 |

blocks, all of which contain the bias term of pair representations. Thus, its speedup is highly related to the fast computation of attention with bias. Since Alphafold 3 is not officially open-sourced, we follow a public high-quality reproduction, Proteinix [8], which is implemented in PyTorch [27].

After a detailed analysis of AlphaFold 3, we find that its efficiency bottleneck is triangle self-attention, which involves the bias matrix projected from intermediate pair representations, making it vary among different samples, layers and heads. To approximate this complex bias, we employ the neural decomposition version of FlashBias, whose inputs $\mathbf{x_q}, \mathbf{x_k}$ are set as the combination of pair and single protein representation. We fine-tune neural basis functions for 10,000 iterations on the PDB dataset. Since we only need to optimize the newly added parameters in $\widehat{\phi}_{\mathbf{q},\theta_1}, \widehat{\phi}_{\mathbf{k},\theta_2}$, this process only takes about 10 hours on a single A100 40GB GPU, then you can infer a new protein with FlashBias. For comparison, the full training process of AlphaFold 3 will take about 7 days on 128 A100 GPUs.

**Results**    Table 6 shows that, compared to the public code, FlashBias can reduce the running time by 32%. In addition, removing bias (*w/o bias*) will significantly improve the efficiency (26.85s vs. 8.27s) but will seriously damage performance. This observation highlights the essentiality of accelerating attention with bias. Despite FlashBias being based on neural decomposition, it will not affect the final performance, whose metric fluctuation is within the standard deviation. Beyond inference, FlashBias can also be a promising tool for training speedup (see Appendix E for details).

**Neural decomposition visualization**    To give a clear illustration of neural decomposition, we also plot the original pair representation bias and FlashBias approximated bias in Figure 7. Neural decomposition can give a relatively accurate estimation of the bias matrix, which performs well in capturing the "texture" of the original bias. In addition, it is also observed that neural decomposition is not completely perfect in reconstructing the diagonal weights. Despite this deficiency, FlashBias still maintains the original accuracy of AlphaFold 3. This may benefit from the dot-product self-attention and residual connection, which can give a robust and dominating weight for relation modeling.

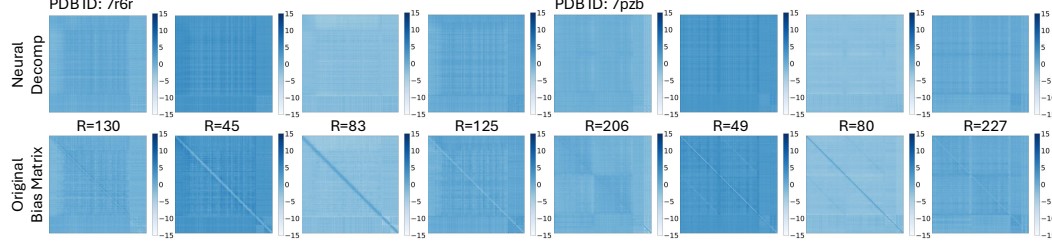

Figure 7: Comparison between neural decomposed factor tensors' multiplication and original bias. Biases of 7r6r (245 residues) and 7pzb (600 residues) in the first layer of Pairformer are plotted, which contains 4 heads. We also mark the rank value that can maintain 99% energy of original biases.

## 5    Conclusions

This paper focuses on the fast computation of attention with bias, which is an essential extension of the attention mechanism and is widely used in language, vision and scientific domains. After an in-depth analysis of FlashAttention, we notice that the optimal efficiency depends on the low-rank property of the attention weight. This further inspires us to present FlashBias based on the low-rank compressed sensing theory, where we also present three practical methods for low-rank decomposition of attention bias, achieving theoretically favorable efficiency. Experimentally, FlashBias can seamlessly support the fast computation of a wide range of famous Transformers without loss of accuracy.

## Acknowledgments and Disclosure of Funding

This work was supported by the National Natural Science Foundation of China (U2342217 and 62021002), the BNRist Innovation Fund (BNR2024RC01010), and the National Engineering Research Center for Big Data Software.

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

# A  Proofs for Theorems in the Main Text

Due to the page limitation of the main text, we present the proofs for theorems in Section 3 here.

**Proof of Theorem 3.1**  This theorem can be proven based on the theoretical analyses in FlashAttention [41] and the basic low-rank compressed sensing knowledge.

*Proof.* According to FlashAttention [41], the theoretical complexity of HBM accesses in the standard attention and FlashAttention are:

$$\text{FlashAttention: } \text{IO}_{\text{flash}} = \Theta\left(\frac{N^2C^2}{S}\right), \text{ StandardAttention: } \text{IO}_{\text{standard}} = \Theta\left(NC + N^2\right), \quad (6)$$

where $\Theta$ represents the asymptotic tight bound. Given $C = \alpha N$ and $S = \beta NC$, the ratio of HBM access in standard attention over FlashAttention is:

$$\frac{\text{IO}_{\text{flash}}}{\text{IO}_{\text{standard}}} = \Theta\left(\frac{(NC + N^2)S}{N^2C^2}\right) = \Theta\left(\frac{(\alpha + 1)\alpha\beta}{\alpha^2}\right) = \Theta\left(\beta(1 + \frac{1}{\alpha})\right). \quad (7)$$

Also, since $R$ is the rank of the attention weight $\mathbf{s}$, which is the dot-product of queries and keys, thus $R \leq C$. Further, we can derive that $\alpha \geq \frac{R}{N}$. $\qquad\square$

**Proof of Theorem 3.2**  As cited in the main text, the least storage cost of an $N \times N$ dense matrix with rank $R$ is equal to $(2NR - R^2)$, whose proof can be found in [6]. Since $R \leq N$, then

$$NR \leq 2NR - R^2 \leq 2NR. \quad (8)$$

Thus, the optimal storage overload for an $N \times N$ matrix is equal to $\Theta(NR)$.

**Proof of Corollary 3.3**  This corollary is derived from Proposition 3. in FlashAttention [10]. Specifically, consider the extreme case that $S = \Theta\left(N(C + R)\right)$, in this case, we have

$$o\left(\frac{NM(C^2 + R^2)}{S}\right) = o\left(\frac{NM(C^2 + R^2)}{N(C + R)}\right) = o\left(M(C + R)\right). \quad (9)$$

Since we have to read the $M \times C$ keys and $M \times R$ decomposed bias factor tensor from HBM, there does not exist an algorithm to finish the computation of attention with bias in $o\left(M(C + R)\right)$ access.

**Proof of Corollary 3.7**  This corollary can be derived from Theorem 2 in FlashAttention [10].

*Proof.* As formalized in Eq. 3, FlashAttention needs to read queries, keys, values, as well as decomposed factor tensors of biases into the on-chip SRAM for computation.

FlashBias's computation is based on the same tiling method as FlashAttention v2 [9]. The $N \neq M$ case corresponds to the cross-attention, which is used for fusing additional information [17], thus we suppose $M \geq N$ here. Suppose that the computation splits queries into $T$ blocks. The overall HBM access complexity is $\Theta\left(N(C + R) + M(C + R)T\right) = \Theta\left(M(C + R)T\right)$. Let the block size of keys and values as $B_{\mathbf{k},\mathbf{v}} \times (C + R)$ and the block size of queries as $B_{\mathbf{q}} \times (C + R)$. Considering the size limitation of SRAM is $S$, following FlashAttention v2 [9], these hyperparameters are set as:

$$B_{\mathbf{q}} = \Theta\left(\frac{S}{C + R}\right), \quad B_{\mathbf{k},\mathbf{v}} = \Theta\left(\min\left(\frac{S}{C + R}, \frac{S}{B_{\mathbf{q}}}\right)\right) = \Theta\left(\min\left(\frac{S}{C + R}, C + R\right)\right). \quad (10)$$

Then, the number of blocks $T = \frac{N}{B_{\mathbf{q}}} = \Theta(\frac{N(C+R)}{S})$. Thus, the overall HBM access complexity is

$$\Theta\left(\frac{NM(C + R)^2}{S}\right) = \Theta\left(\frac{NM(C^2 + R^2)}{S}\right). \quad (11)$$

Recall Corollary 3.3, we can find that the above theoretical complexity is quite well-optimized as there does not exist an algorithm with $o\left(\frac{NM(C^2+R^2)}{S}\right)$ complexity. $\qquad\square$

**Calculation in Example 3.9**  Considering $N, M \gg C, R$, it is easy to calculate the HBM access ratio between FlashAttention and FlashBias as follows:

$$\frac{(1 + \frac{C^2}{S})S}{C^2 + R^2} \approx \frac{(1 + \frac{2\times64^2}{100\times1024})100 \times 1024}{2(64^2 + 64^2)} \approx 6, \quad (12)$$

where we consider the half-precision float, whose storage cost is equal to 2B.

# B  Attention Speed Measurement

As a practical technique, in the main text, we primarily focus on the efficiency benefits to the entire Transformer brought by FlashBias. As a supplement, we conduct a detailed comparison solely on the efficiency of the attention mechanism's computation. Specifically, we compared several advanced implementations of attention: FlashAttention [9], PyTorch SDPA[5] and xFormers [22], as well as the vanilla PyTorch implementation. Since the official implementation of FlashAttention does not support backpropagation for bias matrix, we adopt an open-soured Triton extension for experiments[6].

**Non-Causal Attention**    In this part, we consider an full attention with 4 heads and each head contains 32 channels. As shown in Figure 8, if we only consider the attention speed, FlashBias is over $2\times$ faster than previous best implementation in inference and $1.3\times$ faster in the training phase.

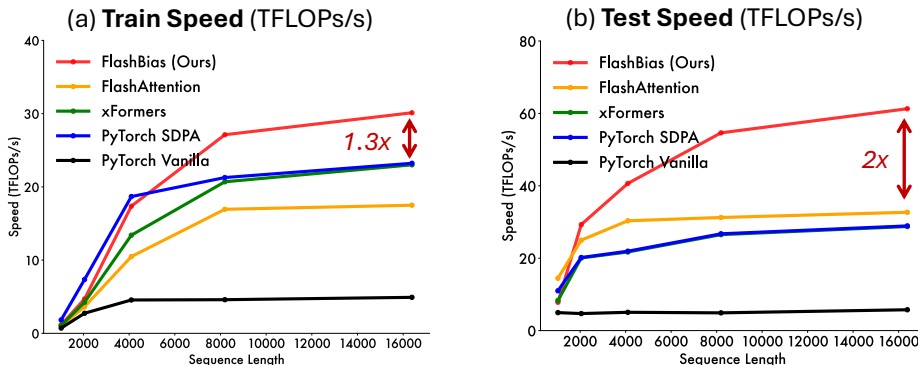

Figure 8: Non-causal attention speed, where attention is with 4 heads and 32 hidden channels per head. $R$ is set as 8. The speed is measured with the batch size as 2.

**Causal Attention**    In this part, we consider the causal attention. Following the experiments in Section 4.2, we configure this attention as a GPT-2 style, which contains 50 heads and 32 channels per head. As presented in Figure 9, FlashBias achieves $1.75\times$ and $1.3\times$ speedup over previous implementations for the inference and training phases, respectively.

It is also notable that all the previous methods suffer from the out-of-memory issue. PyTorch SDPA and xFormer are more memory-efficient but still fail when the sequence length reaches 16,384. This is because they have to save the entire bias matrix with shape of `num_heads`×`seq_len`×`seq_len` in the GPU HBM, which is memory-prohibited in handling extremely long sequences. In contrast, FlashBias utilizes the inherent low-rank property of the bias term, thereby only needing to save the decomposed factors. This experiment also highlights the memory advantage of FlashBias.

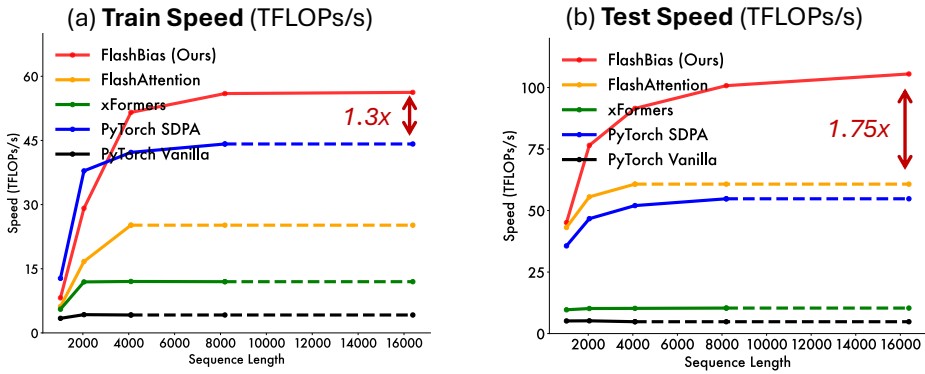

Figure 9: Causal attention speed, where attention is with 50 heads and 32 hidden channels per head. $R$ is set as 2. Dotted lines indicate out-of-memory. Speed is measured with the batch size as 1.

---

[5]PyTorch implementation of `torch.nn.functional.scaled_dot_product_attention`
[6]Triton implementation of FlashAttention with bias.

## C  Speedup of Pangu-Weather [4]

Pangu-Weather [4] is a significant step in adopting Transformers for global weather forecasting. Specifically, its backbone is a 3D Swin Transformer with a hierarchical structure, which contains two different scales. Its speedup is similar to our experiments in Section 4.3. As listed in Table 1, we consider to speedup this model based on the SVD decomposition version of FlashBias.

**Setups**  Since Pangu-Weather is based on the 3D window (with shape $2 \times 6 \times 12$), its bias for relative position encoding is slightly different from Swin Transformer. Especially, its bias is in the shape of $\# \mathrm{num} \times \# \mathrm{heads} \times 144 \times 144$ for each block, where $\# \mathrm{num}$ represents the number of 3D windows. According to meteorological knowledge, different longitudes share the same bias.

**Implementations**  Since Pangu-Weather does not provide accessible model weights or code, our experiments are based on an open-sourced PyTorch reproduction[7]. All the other implementations are the same as the descriptions in Section 4.3. Notably, we find that only relative position biases in the fine scale are low-rank. Thus, we only apply FlashBias to the four 3D Swin layers in the fine scale, where we set $R = 56$ to maintain 99% energy of the original bias matrix. As discussed in Section 4.3, FlexAttention fails in processing such dynamic bias; we didn't compare with it in this task. Since the whole ERA5 data is over 150TB, we only test the model based on 100 samples in 2024.

**Results**  As presented in Table 7, FlashBias can also speedup Pangu-Weather. Compared to the open-sourced code, FlashBias reduces over 20% running time and over 50% GPU memory usage. However, due to the limited sequence length ($N = 144$ in this case), the running time speedup is not as considerable as SwinV2. Just as plotted in Figure 3 and 4, FlashBias will bring more signif-

Table 7: Experiments of Pangu-Weather [4] on ERA5 [15]. Output difference measures of the L2 distance between the outputs of FlashBias and the official code, averaged from 100 different inputs. This difference is calculated on the z-score normalized [28] model outputs to balance diverse meteorological factors.

| Method | Output Difference | Time(s/100iters) | Mem(MB) |
|---|---|---|---|
| Open-sourced Code | - | 98.022 | 26552 |
| FlashAttention w/o bias | 0.0128 | 74.089 | 12141 |
| FlashAttention w/ bias | - | 79.649 | 13186 |
| **FlashBias (Ours)** | 0.0003 | **76.779** | **12222** |

icant speedup in running time and GPU memory for long sequences. As improving the spatial resolution of forecasting is a golden problem in weather prediction [44], we believe that FlashBias has a strong advantage in supporting future research on higher-resolution reanalysis data.

## D  More Results of ALiBi Bias [30]

As we stated in the main text, the released FlashAttention supports an `ALiBi_slopes` feature, where only ALiBi slope values in the shape of $\# \mathrm{heads}$ are loaded from HBM, and the specific bias in the shape of `Block_q`×`Block_k` is created in just-in-time (JIT) compilation[8]. Here `Block_q` and `Block_q` denote the tilling size in FlashAttention [10, 9].

Note that ALiBi is quite a simple bias, and the implementation of FlashAttention's `ALiBi_slopes` feature takes advantage of that the bias values can be created in JIT. In the following experiments, we also utilized this property of ALiBi in FlashBias, where we generate the decomposed

Table 8: Experiments on GPT-2 size model with ALiBi bias. Here we experiment with FlashAttention's `ALiBi_slopes` feature and FlashBias with JIT speedup. The running time for 100 iterations at both training and inference phases when $N = 2048$ are recorded.

| Method | Training Phase (s) | Test Phase (s) |
|---|---|---|
| FlashAttention w/o bias | 119.3 | 38.77 |
| FlashAttention with `ALiBi_slopes` | 119.8 | 38.98 |
| **FlashBias with decomposition in JIT** | 119.8 | 38.98 |

factor tensors of shape `Block_q`×2 and `Block_k`×2 in JIT, instead of directly creating the `Block_q`×`Block_k` matrix. As presented in Table 8, since the difference between FlashAttention and FlashBias is in JIT, these two implementations are almost at the same speed.

---

[7]https://github.com/zhaoshan2/pangu-pytorch
[8]Implementation of `ALiBi_slopes` feature in FlashAttention

# E   More Results in AlphaFold 3 [1]

**Analysis of model efficiency**    The theoretical complexities of triangle self-attention and triangle multiplication in AlphaFold 3 are both cubic w.r.t. the sequence length, while other components: data embedding, single attention with pair bias and feedforward layer, are of linear or quadratic complexity. Thus, according to theoretical complexity, these "triangle operations" are the bottleneck.

We further record the time cost of each component in AlphaFold 3 [1] during the inference of PDB ID 7wux. As listed in Table 9, the triangle attention accounts for 53.3% of the total inference time.

Table 9: Running time of each component in AlphaFold 3 [1] for inferring PDB ID 7wux.

| Component | Complexity w.r.t. Sequence Length | Running Time (s) | Ratio |
|---|---|---|---|
| Data Embedding | Linear | 1.91 | 7.1% |
| **Triangle Self-attention** | **Cubic** | **14.32** | **53.3%** |
| Triangle Multiplication | Cubic | 9.97 | 37.1% |
| Single Attention with Pair Bias | Quadratic | 0.48 | 1.8% |
| FeedForward | Linear | 0.17 | 0.7% |

**Potential in speeding up training of AlphaFold 3 [1]**    In the main text, we only evaluate FlashBias during the inference phase. Going further, as mentioned at the end of Section 3.2, if we directly replace the bias matrix in AlphaFold 3 [1] with two decomposed factor functions, we can also accelerate the training process. As shown in Table 10, FlashBias can save around 15.2% running time and 17.7% GPU memory usage compared to the open-sourced code.

Table 10: Experiments of FlashBias in accelerating the training process of AlphaFold 3 [1], where the default setting is to crop all the residue sequence within 384 tokens.

| Method | Running Time (s / 10 iters) | Maximum GPU Memory (GB) |
|---|---|---|
| Open-sourced Code | 165 | 23.572 |
| FlashAttention with bias | 153 | 23.572 |
| **FlashBias (Ours)** | 140 | 19.390 |

However, since the complete training of AlphaFold 3 will require around 128 GPUs for 7 days, considering the resource limitation, we would like to leave the verification of the final performance of FlashBias-accelerated AlphaFold 3 as our future work.

# F   More Results in Swin Transformer V2 [24]

**Statistics of Swin Transformer V2 bias**
We visualize the rank of bias matrices that can remain 95% energy across 24 layers. As presented in Figure 10, the bias of later layers are of lower rank. To reduce the accumulation error, we only apply Flash-Bias to the last 8 layers as described in Section 4.3. Specifically, for the last 8 layers, we set $R$ in FlashBias as 16. In fact, applying FlashBias to more layers will bring more significant speedup, while to maintain the model accuracy, we only apply Flash-Bias to the last 8 layers in this work.

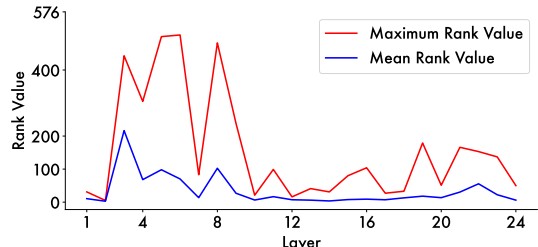

Figure 10: Maximum and mean rank value to remain 95% energy of bias matrices across different heads of 24 layers in SwinV2-B.

**SVD decomposition visualization**    In Figure 6 of the main text, we visualize the original bias matrix at the 20th layer with 16 heads. To deliver an intuitive understanding, we also take this layer's bias as an example and compare it with the SVD decomposed factor tensors' multiplication. As shown in Figure 11, SVD decomposed factor tensors can accurately reconstruct the original bias.

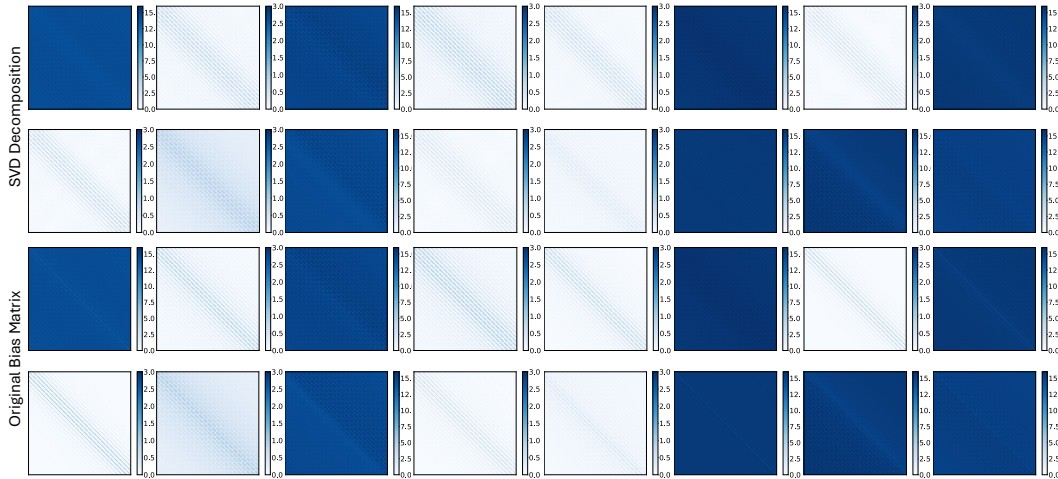

Figure 11: Comparison between SVD decomposed factor tensors' multiplication and original bias.

# G  More Results in Transformer PDE Solver

In Table 5, we only present the efficiency comparison. Further, to demonstrate the performance benefits brought by spatial distance bias, we also include the performance metric in Table 11. With spatial distance bias, the error of the estimated drag coefficient can be reduced by 65.3%, which is a significant progress in industrial design. This further confirms the importance of attention with bias.

Table 11: Performance comparison among attention w/o bias and w/ bias in PDE solving. The relative L2 of surface pressure and surrounding velocity is recorded. We also calculate the drag coefficient $C_D$ based on model-predicted physics fields, whose relative L2 w.r.t. ground truth is also included.

| Method (Sequence Length $N = 32186$) | Surface Pressure Error | Surrounding Velocity Error | $C_D$ Error |
|---|---|---|---|
| Pure attention without spatial distance bias | 0.0838 | 0.0278 | 0.0173 |
| FlashAttention with spatial distance bias | OOM | OOM | OOM |
| FlashBias with spatial distance bias | 0.0706 | 0.0201 | 0.0113 |
| Relative Promotion | 15.7% | 27.7% | 65.3% |

# H  Generalization for Diverse Biases

In FlashBias, we present a neural decomposition version to fit complex and dynamic biases, which has been tested in speeding up AlphaFold 3 in Section 4.4. To further demonstrate the expressive capability of neural decomposition, in this section, we will train neural factor functions $\widehat{\phi}_{\mathbf{q},\theta_1}, \widehat{\phi}_{\mathbf{k},\theta_2}$ to approximate more diverse biases, which can be meaningful for scientific tasks.

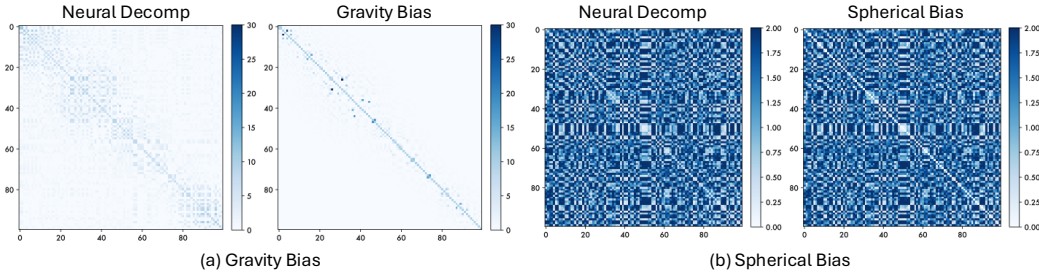

Figure 12: Adopting neural decomposition techniques in FlashBias for more diverse biases.

**Gravity bias**  Many phenomena are inherently governed by underlying physical forces, where gravity is one of the basic factors. Accurately approximating the gravity force is essential for the

modeling of planetary motion [33] or electronic simulation [13]. Thus, we consider introducing the gravity bias into the attention mechanism. Specifically, this bias can be formalized as follows:

$$f(\mathbf{x}_{\mathbf{q},i}, \mathbf{x}_{\mathbf{k},j}) = \frac{1}{\|\mathbf{x}_{\mathbf{q},i} - \mathbf{x}_{\mathbf{k},j}\|_2^2}, \tag{13}$$

where $\mathbf{x}_{\mathbf{q},i}, \mathbf{x}_{\mathbf{k},j}$ denotes the spatial positions of the $i$-th and $j$-th objects. We train $\widehat{\phi}_{\mathbf{q},\theta_1}, \widehat{\phi}_{\mathbf{k},\theta_2}$ based on randomly sampled points from $[0,1] \times [0,1]$ in the 2D space. Since the bias is inversely proportional to spatial distance, we further add 0.01 to the diagonal bias for numerical stability.

**Spherical distance**  When analyzing atmospheric circulation, it is intuitive to consider it as the dynamics on a spherical surface. Therefore, previous research has attempted to introduce spherical Fourier analysis into the model design [5]. Another alternative approach is to add a spherical distance bias to attention in Transformer-based models. Thus, we also consider the spherical distance bias:

$$f(\mathbf{x}_{\mathbf{q},i}, \mathbf{x}_{\mathbf{k},j}) = 2 \cdot \arcsin\left(\sqrt{\sin^2(\frac{\mathbf{x}_{\mathbf{q},i,0} - \mathbf{x}_{\mathbf{k},j,0}}{2}) + \cos \mathbf{x}_{\mathbf{q},i,0} \cos \mathbf{x}_{\mathbf{k},j,0} \sin^2(\frac{\mathbf{x}_{\mathbf{q},i,1} - \mathbf{x}_{\mathbf{k},j,1}}{2})}\right), \tag{14}$$

where $\mathbf{x}_{\mathbf{q},i}, \mathbf{x}_{\mathbf{k},j} \in \mathbb{R}^2$ records the latitude and longitude of the $i$-th and $j$-th position respectively. Similarly, we also train $\widehat{\phi}_{\mathbf{q},\theta_1}, \widehat{\phi}_{\mathbf{k},\theta_2}$ based on randomly sampled points in $[-\pi, \pi] \times [0, 2\pi]$.

For the biases mentioned above, we set $R = 32$ and $\widehat{\phi}_{\mathbf{q},\theta_1}, \widehat{\phi}_{\mathbf{k},\theta_2}$ as three linear layers with in-between tanh activation functions. Then we optimize these model parameters with the Adam [19] optimizer for 10,000 iterations, which will take less than 30 seconds on an A100 GPU. As presented in Figure 12, neural decomposition performs very well in these two biases, especially for the spherical bias. As for gravity bias, since the numerical instability of the inverse proportion function, it is more difficult for optimization, while our method still captures the locality of the bias matrix.

The above experiments further testify the capability of FlashBias in accelerating broader scenarios.

# I  Implementation Details

As a supplement to the main text, we include more details of implementation here. All the experiments are conducted based on PyTorch 2.5.0 [27] and Triton 3.0.0 [38] on a single A100 GPU with 144 CPU cores. All efficiency metrics are averaged from 1,000 iterations. For example, some metrics are recorded as *s/100iters* or *s/10iters*, where we divide the running time of 1,000 iterations by 10 or 100, respectively. In our experiments, all the algorithms' efficiency performance is quite stable.

**ALiBi in Large Language Model**  Since the experiments for ALiBi speedup are based on directly replacing the ALiBi bias with the exact decomposition, FlashBias's output results are completely equal to the original version. The training efficiency is tested under the Adam [19] optimizer.

**Relative position encoding in Swin Transformer**  This part of the experiments strictly follows the official code in Swin Transformer V2 [24]. As illustrated in Figure 10, we only apply FlashBias to the last 8 layers. It is also possible to apply FlashBias to part of low-rank heads in each layer. However, this will require some preprocessing steps, such as rearranging the heads and model parameters to ensure that all low-rank heads are memory continuous. Considering the implementation simplification, we eventually adopt the current design, which only speeds up the final layers.

**Spatial distance bias in PDE solver**  In this experiment, we follow the code base provided here[9], which includes the application of Transformers for the car design task. Specifically, the input contains the position of the pre-defined computation mesh and the output is the pressure and air velocity on these computation mesh points. In this task, we also adopt the exact decomposition for computation. The training phase is also based on the Adam [19] optimizer with a batch size of 1.

**Pair representation bias in AlphaFold 3**  In AlphaFold 3 [1], we adopt the neural decomposition version. This version involves the training of newly added neural layers, whose finetuning configuration is listed in Table 12. Here we find that optimizing the factor functions for 10,000 iterations can obtain a nearly converged version. This may be because the factor functions are token-wise, which

---

[9]https://github.com/thuml/Neural-Solver-Library

Table 12: Configuration for finetuning factor functions $\widehat{\phi}_{\mathbf{q},\theta_1}, \widehat{\phi}_{\mathbf{k},\theta_2}$.

| Part | Configuration |
|------|---------------|
| Input $\mathbf{x_q}, \mathbf{x_k}$ | Sum of row and column in pair representation, with shape of $N \times 128$
Single representation at model beginning with shape of $N \times 449$ |
| Output $\widehat{\phi}_{\mathbf{q},\theta_1}(\mathbf{x_q}), \widehat{\phi}_{\mathbf{q},\theta_1}(\mathbf{x_k})$ | Decomposed factor tensors with shape of $N \times \#\,\text{heads} \times 96$ ($\#\,\text{heads} = 4$) |
| Model $\widehat{\phi}_{\mathbf{q},\theta_1}, \widehat{\phi}_{\mathbf{k},\theta_2}$ | Input Dim: 577; Hidden Dim: 256; Output Dim: 384 ($= 4 \times 96$)
Three linear layers with in-between tanh activation function |
| Training | Initial learning rate: 0.001; Optimizer: Adam;
Learning rate decay: every 50 iterations, reduce to the origin's 0.95
Overall steps: 10,000 iterations |
| Dataset | Train set: "weightedPDB_before2109_wopb_nometalc_0925"
Test set: "recentPDB_1536_sample384_0925" |

means at every iteration, the model will receive $N$ (sequence length) samples for training. To sum up, these layers have been optimized with around 3,840,000 "samples" after 10,000 iterations. Also, to reduce the cumulative error, we only apply FlashBias to the first 16 Pairformer blocks.

## J  Extension to Multiplicative Bias

In this section, we will discuss the extension to multiplicative bias in the following formalization:

$$\mathbf{o} = \text{softmax}(\frac{\mathbf{q}\mathbf{k}^\top}{\sqrt{C}} \odot \mathbf{b})\mathbf{v}, \tag{15}$$

where $\mathbf{b} \in \mathbb{R}^{N \times N}$ represents the multiplicative bias and $C$ denotes the number of hidden channels. $\odot$ denotes the Hadamard product, namely the element-wise multiplication.

**Rotary position embedding (RoPE) [37]**  As mentioned in Section 2.1, RoPE is a special multiplicative bias. Specifically, its pre-softmax attention weight is defined as follows:

$$a_{mn} = \text{Re}[\sum_{i=0}^{C/2-1} \bar{\mathbf{q}}_i \bar{\mathbf{k}}_i e^{i(m-n)\theta_i}], \tag{16}$$

where $a_{mn}$ represents the pre-softmax weight between $m$-th query and $n$-th key. $\bar{\mathbf{q}}_i \in \mathbb{R}^{1 \times 2}$ represents the $2i$ and $2i+1$-th element of the $m$-th query representation. By comparing the above Eq. (15) and Eq. (16), we can obtain the following observations:

*(i) RoPE does not follow the common formalization of multiplicative bias.* As formalized in Eq. (16), RoPE does not directly multiply a bias weight to each attention value, which also involves a detailed reweighing along the channel dimension. Thus, we prefer to consider RoPE as a unique technique, not a "general formalization" for multiplicative bias.

*(ii) RoPE is already FlashAttention-friendly and does not need further modifications to fit the computation of FlashAttention.* According to Eq. (34) in its official paper [37], RoPE can be accomplished by multiplying the rotary tensors with $\mathbf{q}$ and $\mathbf{k}$ before computing $\mathbf{q}\mathbf{k}^\top$. With this design, it does not need to load an additional $N \times N$ bias matrix; thereby, it can be seamlessly integrated with FlashAttention. Thus, we do not think it needs further modifications to fit the computation flow of FlashAttention. That is also why we do not experiment with RoPE in this paper.

**General multiplicative bias**  Here, we will demonstrate that the low-rank decomposition proposed in FlashBias can be extended to multiplicative bias defined in Eq. (15).

Specifically, suppose that $\mathbf{b} \in \mathbb{R}^{N \times N}$ can be decomposed as the multiplication of two rank-$R$ tensors: $\phi_{\mathbf{q}}$ and $\phi_{\mathbf{k}} \in \mathbb{R}^{N \times R}$. We can rewrite the calculation of attention with multiplicative bias as follows:

$$\mathbf{o} = \text{softmax}(\frac{\mathbf{q}\mathbf{k}^\top}{\sqrt{C}} \odot \mathbf{b})\mathbf{v} = \text{softmax}(\frac{\mathbf{q}'\mathbf{k}'^\top}{\sqrt{C}})\mathbf{v},$$

where $\mathbf{q}' = [\mathbf{q} \odot \phi_{\mathbf{q},1}, \cdots, \mathbf{q} \odot \phi_{\mathbf{q},R}] \in \mathbb{R}^{N \times CR}$, $\mathbf{k}' = [\mathbf{k} \odot \phi_{\mathbf{k},1}, \cdots, \mathbf{k} \odot \phi_{\mathbf{k},R}] \in \mathbb{R}^{N \times CR}$.
$$\tag{17}$$

Here $\phi_{\mathbf{q},i}, \phi_{\mathbf{k},i} \in \mathbb{R}^{N \times 1}$ represents the $i$-th channel of the decomposed factor tensors $\phi_{\mathbf{q}}$ and $\phi_{\mathbf{k}}$, whose channel dimension will be broadcasted $C$ times before multiply to $\mathbf{q}$ or $\mathbf{k}$. Operation $[\cdots, \cdots]$ denotes the concatenation along the channel dimension. Besides, the above computation also requires repeating the original $\mathbf{q}$ and $\mathbf{k}$ along the channel dimension for $R$ times and multiplying the factor tensors for each repeat. It is easy to verify the correctness of the above reformalization.

**Example J.1.** *Given the multiplicative bias $\mathbf{b}_{ij} = \cos(i-j)$, which can also be viewed as a simplified version of RoPE (Eq. (16)), $\mathbf{b}$ can be decomposed into two factor tensors with $R = 2$:*

$$\phi_{\mathbf{q}}(\mathbf{x}_{\mathbf{q},i}) = [\cos(i), \sin(i)] \in \mathbb{R}^{1 \times 2}, \quad \phi_{\mathbf{k}}(\mathbf{x}_{\mathbf{k},j}) = [\cos(j), \sin(j)] \in \mathbb{R}^{1 \times 2}. \tag{18}$$

*Based on this decomposition, we can use the above Eq. (17) for speedup, where the whole $N \times N$ bias matrix is transformed into the reweighted repeat of $\mathbf{q}$ and $\mathbf{k}$.*

**Corollary J.2** (Efficiency analysis). *FlashBias's extension to attention with multiplicative bias (Eq. (15)) can reduce the storage complexity of vanilla FlashAttention when $R \leq \sqrt{\frac{S}{C^2} + 1}$.*

*Proof.* According to [9], the HBM access of FlashAttention with bias is $\Theta(\frac{NMC^2}{S} + NM)$, it is easy to drive that the HBM access of Eq. (17) is $\Theta(\frac{NMC^2R^2}{S})$. Thus, when $R \leq \sqrt{\frac{S}{C^2} + 1}$, we have:

$$\Theta(\frac{NMC^2R^2}{S}) \leq \Theta(\frac{NMC^2(\frac{S}{C^2} + 1)}{S}) = \Theta(\frac{NMC^2}{S} + NM). \tag{19}$$

$\square$

**Example J.3** (Design practice). *Given a regular setting $C = 64$ and $S = 100KB$, FlashBias can achieve acceleration for attention with multiplicative bias if there exists a decomposition with $R \leq 27$.*

Since the multiplicative bias is not as common as the additive bias and RoPE has already been well implemented, we would like to leave more explorations of multiplicative bias as our future work.

## K  Limitations

As mentioned in the main text, FlashBias's speedup is based on the low-rank nature of attention bias. Although the low-rank biases are observed in many well-established backbones, if the bias matrix is not of low rank, FlashBias will have to trade off between efficiency and final performance.

About this limitation, one possible solution is to introduce an additional sparsity matrix as a supplement to low-rank decomposition, which is a well-known matrix approximation technique [7]. Specifically, given the attention bias matrix $\mathbf{b}$, this method formalizes the decomposition process as a convex relaxation problem, which can be formalized as follows:

$$\widehat{\mathbf{r}} + \widehat{\mathbf{t}} = \arg\min_{\mathbf{r},\mathbf{t}} \|\mathbf{r}\|_* + \gamma\|\mathbf{t}\|_1,$$
$$\text{s.t. } \mathbf{r} + \mathbf{t} = \mathbf{b}, \tag{20}$$

where $\|\|_*$ represents the *nuclear norm* and $\mathbf{r}$ is expected to be low-rank and $\mathbf{s}$ is optimized to be sparse. Afterwards, the low-rank matrix $\mathbf{r}$ can be sped up through FlashBias, and we can adopt the same technique as attention mask speedup [35] for accelerating the sparse component $\mathbf{t}$. Further, this technique can also make up for the current shortage of FlashBias in approximating the diagonal value (Figure 7), where we can decompose the bias into the sum of a low-rank and a diagonal matrix.

Since we mainly focus on the low-rank property of attention bias in this paper and it will not affect the final performance in our experiments, we would like to leave this as our future work.

## L  Broader Impacts

This paper presents FlashBias as a practical method to enable fast computation of attention with bias, which shows a significant efficiency advantage over FlashAttention and FlexAttention [11]. FlashBias also enables 1.5× speedup for Pairformer in AlphaFold 3 [1] and over 2× speedup in vision and language models without performance drop. These efficiency benefits can facilitate future fine-tuning or development of Transformers in real-world applications. Also, this paper considers a new low-rank compressed sensing perspective in fast attention computation, which can be inspiring for future research. Since we purely focus on the algorithm design for fast computation of attention with bias, there are no potential negative social impacts or ethical risks.

