# OpenReview forum: "FlashBias: Fast Computation of Attention with Bias"
_NeurIPS.cc/2025/Conference — NeurIPS 2025 poster_

### Official Review · Reviewer_UJUf · 2025-06-04

**Clarity:** 3
**Significance:** 3
**Originality:** 3
**Rating:** 5
**Confidence:** 3

**Summary:**

FlashBias addresses the long-standing efficiency gap for attention with bias by leveraging low-rank properties, providing a generalizable, high-performance solution across diverse domains. It seamlessly integrates with existing Transformers, enabling faster training/inference without compromising accuracy, and paves the way for scaling complex bias-aware models.

**Questions:**

- Can the authors provide more comprehensive experiments across a wider variety of architectures and tasks to demonstrate the generality of FlashBias?
- How easy is it to integrate FlashBias into existing model architectures? Are there plans or existing implementations?

**Ethical Concerns:**

["NO or VERY MINOR ethics concerns only"]

**Final Justification:**

The authors effectively addressed key concerns. Expanded experiments (Sandwich bias in GPT-2, PatchTST for time series) plus initial tests on 5 backbones confirm FlashBias’s generality. Clarifications on integration and explanations for limited applicability to Qwen/Llama etc. are reasonable. No critical unresolved issues. Its strengths—efficiency, usability, cross-domain value—outweigh minor gaps, justifying a positive recommendation.

**Limitations:**

Yes.

**Quality:**

4

**Strengths And Weaknesses:**

Strengths
- Rigorous theory links bias rank to GPU efficiency; comprehensive cross-domain experiments validate speedups without accuracy loss.
- Fills a critical gap in bias-aware attention optimization, enabling 1.5×–2× speedups in AlphaFold, Swin Transformer, etc., with real-world utility.
- Novel application of low-rank compressed sensing to attention bias; neural decomposition for dynamic biases offers unique adaptability.

Weaknesses
- While the paper reports significant speedups, it would benefit from a broader evaluation across diverse tasks and models to establish the generality of FlashBias.

---

> ### Author Rebuttal · Authors · 2025-07-29
>
> Many thanks to Reviewer UJUf for providing a detailed review and insightful suggestions.
>
> > **Q1:** "While the paper reports significant speedups, it would benefit from a broader evaluation across diverse tasks and models to establish the generality of FlashBias." "Can the authors provide more comprehensive experiments across a wider variety of architectures and tasks to demonstrate the generality of FlashBias?"
>
> In the original submission, we have tested FlashBias in **5 well-established backbones, such as GPT-2, Swin Transformer, Pangu-Weather and AlphaFold 3, covering language, vision and science**. We think these experiments can provide strong support for the generality of FlashBias.
>
> Following the reviewer's suggestion, we further evaluated FlashBias in speeding up the **Sandwich bias** [1], which was originally proposed for natural language processing. Here are the experiment details.
>
> [1] Chi et al., Dissecting Transformer Length Extrapolation via the Lens of Receptive Field Analysis, ACL 2023
>
> **(1) Sandwich bias:** Specifically, Sandwich bias matrix $\mathbf{b}\in\mathbb{R}^{N\times N}$ is defined as follows:
>
> $$b_{ij}=\sum_{k=1}^{C/2}\cos(\frac{i-j}{10000^{2k/C}}),$$
>
> where $C$ represents the number of channels of deep representation, such as queries, keys and values.
>
> **(2) FlashBias design:** Sandwich bias described above can also be sped up through the "Exact decomposition" version of FlashBias, where the decomposed factor functions $\phi_{\mathbf{q}},\phi_{\mathbf{k}}$ are defined as follows:
>
> $$\phi_{\mathbf{q}}(\mathbf{x}_{\mathbf{q},i})=[\cos(\frac{i}{10000^{2/c}}),\sin(\frac{i}{10000^{2/c}}),\cdots,\cos(\frac{i}{10000^{2(\frac{C}{2})/c}}),\sin(\frac{i}{10000^{2(\frac{C}{2})/c}})]\in\mathbb{R}^{1\times C}$$
>
> $$\phi_{\mathbf{k}}(\mathbf{x}_{\mathbf{k},j})=[\cos(\frac{j}{10000^{2/c}}),\sin(\frac{j}{10000^{2/c}}),\cdots,\cos(\frac{j}{10000^{2(\frac{C}{2})/c}}),\sin(\frac{j}{10000^{2(\frac{C}{2})/c}})]\in\mathbb{R}^{1\times C}.$$
>
> With simple trigonometric function knowledge, it is easy to verify that based on the above definition of factor functions, $\mathbf{b}=\phi_{\mathbf{q}}\phi_{\mathbf{k}}^\top$.
>
> **(3) Efficiency comparison:** The efficiency comparisons are listed below. For this type of attention bias, we also examine the speed advantage of FlashBias in GPT-2. As shown below, FlashBias can also achieve a significant efficiency promotion compared to the vanilla FlashAttention with Bias, which is 23% for training and 17% for inference.
>
> | GPT-2 with Sandwich bias (Sequence Length=8192, using gradient checkpointing) | Train Time (s/100 iters) | Inference Time (s/100 iters) |
> | - | - | - |
> | FlashAttention with Bias                                     | 853.21                   | 196.01                       |
> | FlashBias                                                    | 658.23                   | 163.33                       |
> | Relative Promotion                                           | 23%                      | 17%                          |
>
> > **Q2:** "How easy is it to integrate FlashBias into existing model architectures? Are there plans or existing implementations?"
>
> It is really convenient to implement FlashBias following the formalization in $\underline{\text{Eq. (3) of main text}}$. All the implementation details can be found in $\underline{\text{Appendix G of our paper}}$.
>
> Specifically, the researchers just need to concatenate the decomposed factor functions with queries and keys following $\underline{\text{Eq. (3) of main text}}$, and then the researcher can seamlessly adopt the implementation of FlashAttention. For further speedup, you can also tune the tilling hyperparameter (such as the size of tilling blocks) to achieve better efficiency, while we do not specially tune these hyperparameters during our experiments. Since we cannot add links during the rebuttal phase, we promise to release the code upon publication.

---

> > ### Comment · Reviewer_UJUf · 2025-08-04
> >
> > I agree that the current set of examples is quite substantial. Considering that if this work can be universally applicable, it would indeed be a remarkable achievement. However, this would require more examples. For instance, in terms of model architectures, how does it perform on popular models such as Qwen and Llama? What about its performance in multimodal models, reinforcement learning, and other areas? If there isn't enough time to conduct these experiments, I can understand, and this won't affect my evaluation of the paper.

---

> > > ### Author Response · Authors · 2025-08-05
> > >
> > > Sincerely, thanks for your feedback and further clarification about your questions. It is a great pleasure to hear that you think our experiments are "quite substantial". Here are more discussions about the FlashBias application in other areas.
> > >
> > > **(1) About Qwen and Llama.**
> > >
> > > Since FlashBias focuses on the computation of *attention with bias*, we are only concerned with the attention mechanism in Qwen or Llama, rather than the other architectures. Specifically, Qwen and Llama are both using the attention with the RoPE mechanism, which is a very special mechanism to introduce relative position among tokens.
> > >
> > > As we stated in $\underline{\text{Q1 of response to Reviewer BRmd}}$, RoPE does not follow a common formalization of attention bias and is already FlashAttention-friendly, thereby does not need FlashBias for further speedup.
> > >
> > > Additionally, we would like to emphasize that, prior to FlashBias, researchers were unable to find an efficient method for computing attention with Sandwich bias or ALiBi bias, which may be a reason why all LLMs currently adopt FlashAttention-friendly RoPE instead. Thus, we think **FlashBias poses a potential that LLMs can try other types of relative position embedding.** It is a promising and interesting direction to train Qwen or Llama with ALiBi or Sandwich, but we would like to leave it as future work due to the limited time and resources.
> > >
> > > **(2) About multimodal models, reinforcement learning.**
> > >
> > > About multimodal models, similar to LLMs, most of the current multimodal models adopt attention with RoPE, which is already FlashAttention-friendly.
> > >
> > > As for reinforcement learning, I think our experiments on the aerodynamics simulation task ($\underline{\text{Section 4.4 of main text}}$) can be promising for building a world model. After checking some popular RL papers, such as decision Transformer (NeurIPS 2021 [1]) or diffusion forcing (NeurIPS 2024 [2]), we found that they mainly use pure causal attention without bias or even an RNN. Thus, we think attention with bias may not be a common usage in that area, since they need to process heterogeneous action-observation tokens.
> > >
> > > [1] Decision Transformer: Reinforcement Learning via Sequence Modeling, NeurIPS 2021
> > >
> > > [2] Diffusion Forcing: Next-token Prediction Meets Full-Sequence Diffusion, NeurIPS 2024
> > >
> > > **(3) New experiments about time series forecasting.**
> > >
> > > As for other areas, following your suggestion, we further conduct an experiment on time series forecasting, where we add an ALiBi bias to PatchTST [3] to reflect the temporal order. Note that in the original design of PatchTST, the authors only adopt the vanilla attention without bias.
> > >
> > > | ETTh1 (input 720 time points, predict future 48 points) | MSE       | Training time (s/batch) |
> > > | ------------------------------------------------------- | --------- | ----------------------- |
> > > | vanilla PathTST (without bias, our reproduction)        | 0.338     | 0.02045                 |
> > > | PathTST with ALiBi Bias (speed up with FlashBias)       | **0.321** | 0.02062                 |
> > >
> > > As shown above, attention bias is helpful for time series forecasting and with FlashBias, applying ALiBi will only incur a little cost, demonstrating the significance of FlashBias in broader areas.
> > >
> > > [3] A Time Series is Worth 64 Words: Long-term Forecasting with Transformers, ICLR 2023
> > >
> > > At last, we want to highlight again, beyond the above areas, **FlashBias's efficiency promotion is much more significant in processing complex biases**, such as spatial bias in vision (Swinv2) or scientific knowledge in SciML (AlphaFold3 or Pangu-Weather).
> > >
> > > In the future revision, we will highlight the scope of FlashBias and include the above discussion about potential applications of FlashBias to other areas.

---

> > > > ### Comment · Reviewer_UJUf · 2025-08-05
> > > >
> > > > Thank you for your response. Having also taken into account the other reviewers' information, I have no further questions at this stage.

---

### Official Review · Reviewer_Bp2x · 2025-06-08

**Clarity:** 4
**Significance:** 2
**Originality:** 3
**Rating:** 5
**Confidence:** 4

**Summary:**

This paper addresses a critical and previously overlooked efficiency bottleneck in modern deep learning: the computation of attention mechanisms that include an additive bias term. While attention with bias is essential for the performance of many state-of-the-art models in domains like vision (e.g., Swin Transformer) and biology (e.g., AlphaFold), its efficient computation has been largely unaddressed. The author finds that the primary bottleneck being the quadratic I/O overhead of reading the large, dense bias matrix from GPU memory.

The core contribution of this paper is that the author finds that the attention bias is low-rank. Based on this, they propose a method FlashBias, which performs an offline low-rank decomposition for the bias, and integrates them into the QK computation. FlashBias achieves comparable performance with obvsious decreasing of the computational time and memory.

**Questions:**

Have you performed any experiments to validate that training a model like Swin Transformer from scratch with factorized bias parameters can achieve the same final accuracy as the original model trained with a full-rank bias?

**Ethical Concerns:**

["NO or VERY MINOR ethics concerns only"]

**Final Justification:**

The author's rebuttal is thorough, and the experimental results are solid, prompting me to assign a higher score. However, since Attention Bias is not a universally applicable setting, this remains the only limitation of the paper, preventing me from giving it a strong accept.

**Limitations:**

See weakness part.

**Quality:**

4

**Strengths And Weaknesses:**

[Strength]
1.This paper presents a clear and coherent logical flow—from identifying the limitations, to proposing FlashBias as a solution, and finally validating it with experimental results that strongly support the initial motivation. The overall reasoning is well-structured and complete.
2.The core theoretical insight of the paper is novel and elegant. Instead of brute-force engineering, the authors reframe the I/O problem through the lens of compressed sensing, leading to the key hypothesis that the efficiency is determined by the rank of the bias matrix. This perspective is original and provides a strong theoretical foundation for their method.
3.The proposed method, FlashBias, is simple, effective, and broadly applicable. The evaluation is exceptionally comprehensive and convincing.

[Weakness]
1.The paper's scope might seem limited as attention bias is not universally used.
2.While the paper shows minimal performance loss at the chosen rank R, the inherent trade-off between rank, approximation error, and model accuracy warrants a more systematic analysis. Such a study would enhance the robustness of the results and offer clearer guidance for applying FlashBias in practice.
3.The paper shows strong inference-time speedups, but training-time benefits are less validated. For example, while factorizing bias terms in models like Swin Transformer is promising, there are no results showing that training from scratch achieves similar accuracy, leaving training-time gains uncertain.

---

> ### Author Rebuttal · Authors · 2025-07-29
>
> Many thanks to Reviewer Bp2x for providing the insightful review and questions.
>
> > **Q1:** "The paper's scope might seem limited as attention bias is not universally used."
>
> We agree with the reviewer's opinion that "attention bias is not universally used in previous research". But we would also want to point out that **the poor efficiency in computing attention with bias may be one of the reasons for this not-universally-used situation.**
>
> As presented in our experiments of "**Transformer-based PDE solvers** ($\underline{\text{Table 5 and Table 9}}$)", spatial bias is significantly helpful in PDE solving, while to our best knowledge, none of the previous Transformer-based neural PDE solvers adopt this important bias. This may be because directly incorporating spatial bias into attention will cause out-of-memory error as presented in $\underline{\text{Table 5 of main text}}$. We believe that FlashBias can promote the future usage of attention bias, which has already been proven useful in many tasks.
>
> > **Q2:** "While the paper shows minimal performance loss at the chosen rank R, the inherent trade-off between rank, approximation error, and model accuracy warrants a more systematic analysis. Such a study would enhance the robustness of the results and offer clearer guidance for applying FlashBias in practice."
>
> Thanks for this valuable suggestion.
>
> **(1) Rank $R$ represents the lower bound that will not lose accuracy.**
>
> Since the **basic principle** in building fast computation methods of attention mechanism is to maintain its original performance, all the experiments in our original submission are based on the spirit of "do not affect the model accuracy".
>
> Thus, we prefer to view the "rank" $R$ used in our paper as the lower bound that does not affect accuracy, not a hyperparameter that needs to be tuned for trading off performance and efficiency.
>
> **(2) More experiments about the performance-efficiency trade-off.**
>
> Following the reviewer's suggestion, we also tested FlashBias in SwinV2 with different rank values. Note that in our experiments, we only adopt FlashBias to **half of the attention heads** in SwinV2 to maintain 95% energy of bias matrices. More implementation details can be found in $\underline{\text{Appendix D}}$.
>
> As shown below, decreasing $R$ will cause a consistent performance drop but bring efficiency advantages. Note that, for the $R=16$ case, as shown in $\underline{\text{Figure 7 in Appendix D}}$, this setting will cause some of the heads to not maintain 95% energy, thereby causing a more serious performance drop. However, even in this setting, FlashBias still achieves better accuracy than quantization techniques INT8 PTQ and brings a more significant speedup (over 60%) over their official code.
>
> | SwinV2                                                       | Running Time (s/batch)                | Top1 Acc |
> | - | - | - |
> | Official Code                                                | 0.473                                 | 87.144%  |
> | FlashBias R=96                                               | 0.210                                 | 87.142%  |
> | FlashBias R=64                                               | 0.198                                 | 87.137%  |
> | FlashBias R=32 (Default setting, see $\underline{\text{Appendix D}}$) | 0.190                                 | 87.126%  |
> | FlashBias R=16 (Cannot consistently maintain 95% energy)     | 0.188                                 | 86.982%  |
> | INT8 PTQ                                                     | Around 22% speed up over official code | 86.46%   |
>
> > **Q3:** "The paper shows strong inference-time speedups, but training-time benefits are less validated. For example, while factorizing bias terms in models like Swin Transformer is promising, there are no results showing that training from scratch achieves similar accuracy, leaving training-time gains uncertain." "Have you performed any experiments to validate that training a model like Swin Transformer from scratch with factorized bias parameters?"
>
> Thanks for this insightful question.
>
> **(1) Clarify the verified effective scope of FlashBias.**
>
> First, we want to highlight that **FlashBias attempts to ensure exact or nearly-exact computation of attention with bias.** Since in SVD and neural decomposition, "replacing bias with factorized parameters" will change the model architecture, which will also affect the backward properties of the model, we only test the inference speedups of SwinV2 and AlphaFold 3, as stated in $\underline{\text{Table 2 of main text}}$. Specifically, as discussed in $\underline{\text{Lines 205-212 of main text}}$, the verified effective scope of FlashBias can be summarized below. **Note that speeding up the inference of widely deployed models (e.g. Swin Transformer and AlphaFold 3) is already very significant.**
>
> | Decomposition | Verified Effective Scope |
> | - | - |
> | Exact         | Training & Inference |
> | SVD           | Inference            |
> | Neural        | Inference            |
>
> For clarity, we will highlight that **the effectiveness of SVD and neural decomposition for the training phase is under exploration** in the $\underline{\text{Limitation and Future Work}}$ section of the revised paper.
>
> **(2) New experiments on training Swin V2 with factorized bias.**
>
> Since both SwinV2 Large and AlphaFold 3 experimented in our paper will take massive computation costs (around 32 A100 GPUs for several days), **we made a quick test on training SwinV1 Tiny, whose training costs 8 GPUs for 2 days.** Here are the results. Since the sequence length is quite small (seq_len = 49), the efficiency advantage is not significant, but this experiment can be used to test the training performance of FlashBias. Following our findings in $\underline{\text{Table 7 of Appendix D}}$, we only replace biases in half of the attention heads at each layer with low-rank factors.
>
> As shown below, training with low-rank bias will not affect the final performance, where the Top1 Acc difference (0.09%) is under the standard deviation (*note: the Top 1 Acc difference on ImageNet within 0.2% is usually treated as normal performance fluctuation*).
>
> Due to the time and resource limitations, we cannot make more large-scale experiments at the rebuttal stage. Considering the scope of this paper, we would like to leave the exploration of FlashBias training performance as our future work.
>
> | SwinV1-Tiny (Attn `seq_len = 49`) | Top1 ACC | Training Time | Training Memory |
> | - | - | - | - |
> | Official Code | 81.29 | 0.3372        | 9191MB          |
> | FlashBias (R=8)                 | 81.20 | 0.3303        | 8892MB          |

---

> > ### Comment · Reviewer_Bp2x · 2025-08-05
> >
> > Thank you for your rebuttal, which addressed several of my concerns. I recognize the solid performance of FlashBias and its robust R selection. While I no longer have any significant doubts, I believe the true value of this work may become evident only when applied to a general large-scale model.

---

> > > ### Author Response · Authors · 2025-08-05
> > > **Thanks for your response**
> > >
> > > We would like to thank Reviewer Bp2x for the inspiring review and valuable feedback.
> > >
> > > We agree that the value of FlashBias is more evident in "general large-scale model". However, we also want to highlight that beyond "general" models, **scientific large-scale models** also deeply affect our daily lives, such as:
> > >
> > > -  **AlphaFold 3 ($\underline{\text{Section 4.4 of main text}}$):** representative work of the Nobel Prize in Chemistry 2025, widely used in biopharmaceutical factories,
> > > - **Pangu-Weather ($\underline{\text{Appendix B of main text}}$):** published in Nature 2023, deployed in the European Center to generate daily weather forecasting.
> > >
> > > And in such scientific areas, attention with bias is much more commonly used, in which scope we think FlashBias can be more "universally" effective.
> > >
> > > Again, thanks for your dedication to our work.

---

### Official Review · Reviewer_BRmd · 2025-07-03

**Clarity:** 3
**Significance:** 3
**Originality:** 3
**Rating:** 5
**Confidence:** 4

**Summary:**

An additive bias term of attention is widely used to guide the learning process. This paper improves the efficiency of bias computation by realizing its low rank nature. Specifically, one doesn't need to store an $N \times N$ bias matrix. Instead, the bias matrix is represented by the product of two $N \times r$ matrices. Each element of the bias matrix is computed in-place of GPU high bandwidth memory which is similar to the characteristic of FlashAttention. Thus, the proposed method is called FlashBias. FlashBias achieves 1.5x speedup for AlphaFold, and over 2x speedup for attention with bias in vision and language models without loss of accuracy.

**Questions:**

In figure 3 and figure 4, which kind of bias is used?

**Ethical Concerns:**

["NO or VERY MINOR ethics concerns only"]

**Final Justification:**

All of my concerns have been addressed. So I improve my score from 4 to 5. First, the authors provide detailed analysis about RoPE and they show that FlashBias can be extended to multiplicative bias. Second, the authors update the experiments about alibi bias and they add `ALiBi_slopes` feature for FlashAttention.

**Limitations:**

The proposed method is not applicable to non-additive bias, such as rotary position embedding (RoPE).

**Quality:**

3

**Strengths And Weaknesses:**

Strengths:
1. The proposed method is easy to implement and achieves significant speedup for attention with bias.
2. The proposed method is verified by various tasks with various bias types, including spatial bias, alibi bias, and pair bias. This paper designs different strategies to obtain the low-rank representations of bias.
3. This paper provides theoretical analysis of the IO complexity of bias. Its efficiency is supported by the theoretical results.

Weaknesses:
1. If my understanding is right, this method is applicable to additive bias but it doesn't work for non-additive bias, such as rotary position embedding (RoPE). However, the authors claim that the proposed method provide a fast-accurate approximation for biases in general formalization. It is better to discuss whether RoPE can be approximated by low-rank representations.
2. The results of alibi bias, i.e. table 3 are not convincing. To my knowledge, the alibi bias has been optimized by FlashAttention in a similar way (although they doesn't formalize alibi bias as a low-rank representation). I believe that the improvements in table 3 have other reasons.

---

> ### Author Rebuttal · Authors · 2025-07-29
>
> We would like to sincerely thank Reviewer BRmd for providing valuable feedback and questions.
>
> > **Q1:** "This method is applicable to additive bias, but it doesn't work for non-additive bias, such as rotary position embedding (RoPE)." "However, the authors claim that the proposed method provides a fast-accurate approximation for biases in general formalization. It is better to discuss whether RoPE can be approximated by low-rank representations."
>
> Thanks for the reviewer's thoughtful question. We will revise "a fast-accurate approximation for biases in general formalization" as **"for additive biases in general formalization"** in the future revision for scientific rigor.
>
> Next, following the reviewer's comments, we would like to discuss more about RoPE and multiplicative bias in the following formalizations:
>
> $$\mathbf{o}=\operatorname{softmax}(\frac{\mathbf{q}\mathbf{k}^\top}{\sqrt{C}}\mathbf{b})\mathbf{v},\ \ \ (1)$$
>
> where $\mathbf{b}\in\mathbb{R}^{N\times N}$ represents the multiplicative bias and $C$ denotes the number of hidden channels.
>
> **(1) RoPE does not follow the formalization in the above Eq. (1) and is already FlashAttention-friendly.**
>
> Our paper identifies the efficiency bottleneck in computing attention with additive bias as: **they have to additionally load a $N\times N$ bias matrix from HBM to SRAM**. However, in RoPE, according to their paper, its pre-softmax attention weight is defined as follows:
>
> $$a_{mn}=\text{Re}[\sum_{i=0}^{C/2-1}\bar{\mathbf{q}}_i\bar{\mathbf{k}}_ie^{i(m-n)\theta_i}],\ \ (2)$$
>
> where $a_{mn}$ represents the pre-softmax weight between $m$-th query and $n$-th key. $\bar{\mathbf{q}}_i\in\mathbb{R}^{1\times 2}$ represents the $2i$ and $2i+1$-th element of the $m$-th query representation.
>
> By comparing the above Eq.(1) and Eq.(2), we can obtain the following observations:
>
> - **RoPE does not follow the common formalization of multiplicative bias ($\underline{\text{Eq. (1) above}}$).** As formalized in $\underline{\text{above Eq. (2)}}$, RoPE does not directly multiply a bias weight to each attention value, which also involves a detailed reweighing along the channel dimension. Thus, we prefer to consider RoPE as a unique technique, not a "general formalization" for multiplicative bias.
> - **RoPE is already FlashAttention-friendly and does not need further modifications to fit the computation of FlashAttention.** According to $\underline{\text{Eq.(34) in their paper}}$, RoPE can be accomplished by multiplying the rotary tensors with $\mathbf{q}$ and $\mathbf{k}$ before computing $\mathbf{q}\mathbf{k}^\top$. With this design, it does not need to load an additional $N\times N$ bias matrix; thereby, it can be seamlessly integrated with FlashAttention. Thus, we do not think it needs further modifications to fit FlashAttention.
>
> Considering the special definition and native efficiency of RoPE, we did not discuss it in the original submission, but will add the above discussion as a separate section in a future revision.
>
> **(2) FlashBias can be extended to multiplicative bias formalized in the above Eq. (1).**
>
> Next, we will show that the low-rank decomposition proposed in FlashBias can be extended to multiplicative bias defined in $\underline{\text{Eq. (1) formalized above}}$.
>
> **Overall design:** Specifically, suppose that $\mathbf{b}\in\mathbb{R}^{N\times N}$ can be decomposed as the multiplication of two rank-$R$ tensors: $\phi_{\mathbf{q}}$ and $\phi_{\mathbf{k}}\in\mathbb{R}^{N\times R}.$ We can rewrite the calculation of attention with multiplicative bias as follows:
>
> $$\operatorname{softmax}(\frac{\mathbf{q}\mathbf{k}^\top}{\sqrt{C}}\mathbf{b})\mathbf{v}=\operatorname{softmax}(\frac{\mathbf{q}^\prime{\mathbf{k}^\prime}^\top}{\sqrt{C}})\mathbf{v},$$
>
> $\text{where}\ \mathbf{q}^\prime=\sqrt{C}[\mathbf{q}\phi_{\mathbf{q},1},\cdots,\mathbf{q}\phi_{\mathbf{q},R}]\in\mathbb{R}^{N\times CR},\  \mathbf{k}^\prime=[\mathbf{k}\phi_{\mathbf{k},1},\cdots,\mathbf{k}\phi_{\mathbf{k},R}]\in\mathbb{R}^{N\times CR}. \ \ \ (3)$
>
> $\phi_{\mathbf{q},i},\phi_{\mathbf{k},i}\in\mathbb{R}^{N\times 1}$ represents the $i$-th channel of $\phi_{\mathbf{q}}$ and $\phi_{\mathbf{k}}$.
> The above computation requires repeating the original $\mathbf{q}$ and $\mathbf{k}$ along the channel dimension for $R$ times and multiplying the factor tensors for each repeat. It is easy to verify the correctness of the above reformalization. And, the above formalization can reduce storage complexity of vanilla multiplicative bias when $R\leq 1+\frac{N}{2C}$.
>
> **Example:** Given the multiplicative bias $\mathbf{b}_{ij}=\cos(i-j)$, which can also be viewed as a simplified version of RoPE, $\mathbf{b}$ can be decomposed as $R=2$ with
>
> $$\phi_{\mathbf{q}}(\mathbf{x}_{\mathbf{q},i})=[\cos(i), \sin(i)]\in\mathbb{R}^{1\times 2}$$
>
> $$\phi_{\mathbf{k}}(\mathbf{x}_{\mathbf{k},j})=[\cos(j), \sin(j)]\in\mathbb{R}^{1\times 2}.$$
>
>  Thus, we can use the above $\underline{\text{Eq. (3)}}$ for speedup, where the whole $N\times N$ bias matrix is transformed into the reweighted repeat of $\mathbf{q}$ and $\mathbf{k}$.
>
> Since the multiplicative bias is not as common as the additive bias and RoPE has already been well implemented, we would like to leave more explorations as our future work.
>
> > **Q2:** "The results of alibi bias, i.e. table 3 are not convincing. To my knowledge, the alibi bias has been optimized by FlashAttention in a similar way (although they doesn't formalize alibi bias as a low-rank representation). I believe that the improvements in table 3 have other reasons."
>
> Thanks for the detailed review and insightful question.
>
> **(1) Clarification about Table 3.**
>
> After carefully checking the implementation of FlashAttention and contacting their authors, we find that our experiments in ALiBi are misled by their official Triton implementation examples. During our initial submission, we found that the official Triton implementation example of FlashAttention with ALiBi is wrong (see Line 493 in `flash-attention/blob/main/flash_attn/flash_attn_triton.py` of the official GitHub repository; they should not directly broadcast the bias vector to the matrix). Thus, we mistakenly thought the current FlashAttention could not support ALiBi well.
>
> This drove us to avoid using the ALiBi speedup feature in FlashAttention. Therefore, the FlashAttention results in Table 3 are still based on the full matrix-type bias. That is why FlashBias can reduce the time for processing bias by around 50%.
>
> **(2) New comparison with FlashAttention and FlashBias.**
>
> Thanks for the reviewer's professional reminder. We find that the officially released FlashAttention is not like their Triton examples. Specifically, the released FlashAttention supports an `ALiBi_slopes` feature, where only ALiBi slope values in the shape of #$\operatorname{heads}$ are loaded from HBM, and the specific bias in the shape of Block_q $\times$ Block_k is created in Jit. Here Block_q and Block_k denote the tilling size in FlashAttention.
>
> Note that ALiBi is quite a simple bias, and the implementation of FlashAttention's `ALiBi_slopes` feature takes advantage of that the bias values can be created in Jit. In the following experiments, we also utilized this property of ALiBi in FlashBias, where **we generate the decomposed factor tensors of shape Block_q$\times$2 and Block_k$\times$2 in Jit**, instead of directly creating the Block_q $\times$ Block_k matrix. However, since the difference between FlashAttention and FlashBias is in Jit, these two implementations are almost at the same speed.
>
> |                                            | Training (s) | Inference (s) |
> | - | - | - |
> | FlashAttention without Bias                | 119.3        | 38.77         |
> | FlashAttention with `ALiBi_slopes` feature | 119.8        | 38.98         |
> | FlashBias with decomposition in Jit        | 119.8        | 38.98         |
>
> For clarity, we will include all the above discussions and the specific usage of FlashAttention in the revised paper.
>
> At last, we would like to highlight that beyond ALiBi, FlashBias is much more useful in processing complex biases, such as SwinV2 or AlphaFold 3. We believe this generality of FlashBias is meaningful to the research community.
>
> > **Q3:** "In figure 3 and figure 4, which kind of bias is used?"
>
> As stated in $\underline{\text{Line 219 of main text}}$, Figures 3 and 4 are just for an "overall efficiency comparison". Thus, these experiments do not relate to any specific type of bias. These experiments are used to compare the model efficiency between loading a #$\operatorname{heads}\times N\times N$ bias matrix and only loading two factor vectors of shape #$\operatorname{heads}\times N\times R$.

---

> ### Author Response · Authors · 2025-08-09
> **Thanks for your acknowledgement and positive review of our paper**
>
> Dear Reviewer BRmd,
>
> Thanks so much for your detailed and professional review. Your initial positive review of our paper means a lot to us.
>
> During the rebuttal stage, following your suggestion, we have further discussed $\underline{\text{RoPE and FlashBias's extension to multiplicative bias}}$, as well as provided clarifications on the $\underline{\text{Table 3}}$ experiments. We sincerely appreciate your valuable questions, which help us a lot in promoting our paper for scientific rigor. We promise to include all the rebuttal discussions in the future revision.
>
> As you have posted an acknowledgement, we think you have made a well-considered final justification. **Hope our rebuttal can resolve your concern to your satisfaction. And we are happy to answer any further questions.** We believe that FlashBias can be a promising technique to support fast computation of complex biases, which is really meaningful for large-scale scientific models, such as **AlphaFold 3, Pangu-Weather, or neural PDE solvers**, as demonstrated in our paper. Your and other reviewers' reviews and feedback make this method more rigorous and generalizable.
>
> Thanks again for your time and engagement.
>
> Authors

---

### Official Review · Reviewer_xU4x · 2025-07-03

**Clarity:** 3
**Significance:** 3
**Originality:** 3
**Rating:** 5
**Confidence:** 3

**Summary:**

This paper addresses the inefficiency of computing attention mechanisms with additive bias terms, which are essential for performance but suffer from quadratic I/O complexity in existing methods like FlashAttention. The proposed method based on low-rank compressed sensing that decomposes large bias matrices into smaller factor tensors. It uses exact decomposition for known low-rank biases, SVD for fixed learned biases, and neural decomposition for complex dynamic biases. Experimental results showed their method achieves substantial speedups—1.5× for AlphaFold and over 2× for vision/language models—while maintaining accuracy. It also drastically reduces GPU memory usage, enabling training on larger-scale cases.

**Questions:**

1. For dynamic biases (Section 3.3), what is the latency/memory cost of the lightweight networks during inference?
2. In mainstream models, which bias matrices are not of low rank? A study on corner cases would be beneficial.
3. Could FlashBias dynamically adjust the decomposition rank \(R\) per layer/head (e.g., based on singular value thresholds)?

**Ethical Concerns:**

["NO or VERY MINOR ethics concerns only"]

**Final Justification:**

The Flashbias method shows significant speedups and memory savings in AlphaFold, Swin Transformer, and GPT-2. The paper is well-organized, clearly written, and supported by strong data. Recommended for acceptance.

**Limitations:**

The authors have discussed most of the limitations of their work and provide the potential solutions. However, it would be beneficial to include more examples or studies related to bias matrices that are not of low rank.

**Paper Formatting Concerns:**

No formatting Concerns have been identified in the paper.

**Quality:**

3

**Strengths And Weaknesses:**

Strengths:
1. Proposed Flashbias provides 1.5–2× speedups in AlphaFold, Swin Transformer, and GPT-2 with ALiBi as well as memory saving.
2. The paper is well-organized and clearly written. The figures and tables are illustrative and effectively support the overall narrative.
3. Focus on the speedup of LLM with bias and presented a novel method.

Weaknesses:
1. Speedups are reported, but energy consumption (e.g., TDP reduction on A100) is unmeasured.
2. In Section 4.4, "After a detailed analysis of AlphaFold 3, we find that its efficiency bottleneck is triangle self-attention," there is no data to support this claim.
2. Some of the images are too small to be easily readable.

---

> ### Author Rebuttal · Authors · 2025-07-29
>
> We would like to sincerely thank Reviewer xU4x for providing a detailed review and insightful questions.
>
> > **Q1:** "Speedups are reported, but energy consumption (e.g., TDP reduction on A100) is unmeasured."
>
> Following the reviewer's suggestion, we further tested the inference power cost of Swin Transformer V2. Since the TDP (Thermal Design Power) is usually predefined in the A100, which is 400W in our device, we record the power draw during experiments.
>
> As shown below, compared to the official code of Swin V2, FlashBias can slightly reduce the average and maximum power draw. Benefitting from a significantly faster running speed, **FlashBias can reduce power costs by 61% (123.6$\to$48.2) per batch.**
>
> | SwinV2 Inference | Avg Power Draw (W) | Max Power Draw (W) | Running Time (s/batch) | Power Cost per Batch (J) |
> | - | - | - | - | - |
> | Original Code    | 261.3              | 290.2              | 0.473                  | 123.6                    |
> | FlashBias        | **253.5**          | **285.4**          | **0.190**              | **48.2**                 |
>
> > **Q2:** There is no data to support "After a detailed analysis of AlphaFold 3, we find that its efficiency bottleneck is triangle self-attention."
>
> The theoretical complexities of triangle self-attention and triangle multiplication in AlphaFold 3 are both cubic w.r.t. the sequence length, while other components are of linear or quadratic complexity. Thus, in the theoretical complexity aspect, these "triangle operations" are the efficiency bottleneck.
>
> As per your request, we have further recorded the time cost of each component in AlphaFold 3 during the inference of PDB ID 7wux. As shown below, the **triangle self-attention accounts for 53.5% of the total inference time.** For scientific rigor, we will revise this statement as *"its efficiency bottleneck is 'triangle operations', where triangle self-attention is one of them."*
>
> | Vanilla Pairformer's Components | Theoretical Complexity w.r.t. Sequence length | Running Time (s) | Ratio of Whole Model |
> | - | - | - | - |
> | Data Embedding                  | Linear                                        | 2.26             | 7.2% |
> | **Triangle Self-attention**     | **Cubic**                                     | **16.83**        | **53.5%** |
> | Triangle Multiplication         | Cubic                                         | 11.68            | 37.1% |
> | Single Attention with Pair Bias | Quadratic                                     | 0.52             | 1.6% |
> | FeedForward                     | Linear                                        | 0.19             | 0.6% |
>
> > **Q3:** "Some of the images are too small to be easily readable."
>
> Thanks for the reviewer's valuable suggestion. Since we cannot update the submission during the rebuttal stage, we will enlarge $\underline{\text{Figs 3 and 4}}$ and the color bar for figures in the Appendix in the future revision.
>
> > **Q4:** "For dynamic biases, what is the latency/memory cost of the lightweight networks during inference?"
>
> Since "lightweight networks" used in FlashBias are just simple linear layers, their computation are quite efficient.
>
> For example, experiments in AlphaFold 3 ($\underline{\text{Table 6  of main text}}$) are based on learnable lightweight networks. Even though FlashBias adds two newly linear layers to each Pairformer block, our method still enables a 1.5x speedup over the open-sourced code. Specifically, for the inference of PDB ID 7wux, the newly added "lightweight networks" will only account for around 0.2% (0.0417s vs. 20.89s) of the total running time and 3.5% of the total memory cost.
>
> | Inferring PDB ID 7wux             | Time (s) | Mem (GB) |
> | - | - | - |
> | Open-Sourced Code                 | 31.48    | 21.58    |
> | FlashBias                         | 20.89    | 10.37    |
> | Lightweight Networks in FlashBias | 0.0417   | 0.37     |
>
> > **Q5:** "In mainstream models, which bias matrices are not of low rank? A study on corner cases would be beneficial." "However, it would be beneficial to include more examples or studies related to bias matrices that are not of low rank."
>
> As presented in $\underline{\text{Fig. 7 of Appendix D}}$, not all the relative position bias matrices in SwinV2 are of low rank, but we can select the low-rank layers or heads and apply FlashBias on these low-rank layers/heads for speedup.
>
> Besides, we also want to highlight that, as proved by $\underline{\text{Theorem 3.2}}$, the rank of the bias matrix determines the optimal storage complexity. Thus, FlashBias does not attempt to speed up all the mainstream models by low-rank decomposition, but presents a possible way to reach the optimal efficiency when the bias matrix presents a low-rank property. We will include these discussions in the $\underline{\text{Limitation section in Appendix H}}$.
>
> > **Q6:** "Could FlashBias dynamically adjust the decomposition rank (R) per layer/head (e.g., based on singular value thresholds)?"
>
> For the SVD decomposition version, the rank can be adjusted for different layers/heads. For instance, in SwinV2 experiments, the selected rank values of different layers and heads are different, as stated in $\underline{\text{Appendix D}}$.
>
> If you refer to the dynamic biases, such as AlphaFold 3, I do not think online SVD is an efficient design, since we need to compute SVD for every inference, as stated in $\underline{\text{Lines 184-186 of main text}}$. That is why we present the neural decomposition version in FlashBias for dynamic biases.

---

> > ### Comment · Reviewer_xU4x · 2025-08-08
> >
> > Thanks to the efforts of the authors. It has addressed most of my concerns. I will keep my decision.

---

> > > ### Author Response · Authors · 2025-08-09
> > > **Thanks for your response and positive support for our paper**
> > >
> > > Thank you very much for your feedback.
> > >
> > > We sincerely appreciate your engagement and support in strengthening the paper. We will include all your suggestions and new experiments in the future revision.

---

### Note · Authors · 2025-08-12

Dear Area Chair and Reviewers,

Thank you for your engagement and valuable feedback. We hope that we have addressed your concerns during the rebuttal and discussion stages. Here we want to summarize and emphasize some key aspects:

**Our contribution:** In this paper, we presented FlashBias for the fast computation of attention with bias, which enables 1.5x speedup for AlphaFold, and over 2x speedup for attention with bias in vision and language models without loss of accuracy. This paper has received consistent positive initial reviews from 4 reviewers. Especially, the novelty and contribution of FlashBias are appreciated by all the reviewers:

- xU4x: "The paper is well-organized and clearly written", "presented a **novel** method".
- BRmd: "The proposed method is easy to implement and achieves **significant speedup**" and "is verified by various tasks with various bias types".
- Bp2x: "This paper addresses a critical and previously overlooked efficiency bottleneck", "The core theoretical insight of the paper is **novel and elegant**", "FlashBias is **simple, effective, and broadly applicable**" and "The evaluation is **exceptionally comprehensive and convincing**."
- UJUf: "FlashBias **addresses the long-standing efficiency gap and fills a critical gap**," and "provides a **generalizable, high-performance** solution."

**Summary of Rebuttal:** During rebuttal, following the insightful questions from reviewers, we further improved this work in the following aspects:

- xU4x: Provided a detailed efficiency analysis for each component of AlphaFold 3.
- BRmd, UJUf: Clarified the relation w.r.t. RoPE and presented the extension of FlashBias to multiplicative bias.
- Bp2x: Provided the performance-efficiency trade-off under various rank choices and adopted FlashBias to speed up the Swin Transformer's training.
- UJUf: Extending FlashBias to new Sandwich bias and time series forecasting task.

We deeply thank all the reviewers and the ACs for their effort in reviewing our paper, which has helped us significantly in improving the experiments and scientific rigor of our work.

We hope the clarifications provided throughout this discussion, along with the strong empirical evidence and contribution acknowledged by the reviewers, demonstrate the merit and value of our work.

We thank you again for your time and consideration.

---

### Decision · Program_Chairs · 2025-09-17

**Decision:**

Accept (poster)

**Comment:**

This paper introduces a method for faster attention computation when additive bias terms are involved. The reviews are uniformly positive, praising its simplicity, strong speedups across a variety of tasks, and GPU memory improvements. There are some remaining doubts on the universality of the application, although they rely on bias not being used in all models, which as the authors note may itself be caused by the inefficiencies being ameliorated in this work. Given the positive reviews and diversity of evaluation I recommend acceptance without reservations.